# Dimensionality Reduction for Representing the Knowledge of Probabilistic Models

**Marc T. Law & Jake Snell**
University of Toronto, Canada
Vector Institute, Canada

**Amir-massoud Farahmand**
Vector Institute, Canada

**Raquel Urtasun**
University of Toronto, Canada
Vector Institute, Canada
Uber ATG, Canada

**Richard S. Zemel**
University of Toronto, Canada
Vector Institute, Canada
CIFAR Senior Fellow

## Abstract

Most deep learning models rely on expressive high-dimensional representations to achieve good performance on tasks such as classification. However, the high dimensionality of these representations makes them difficult to interpret and prone to over-fitting. We propose a simple, intuitive and scalable dimension reduction framework that takes into account the soft probabilistic interpretation of standard deep models for classification. When applying our framework to visualization, our representations more accurately reflect inter-class distances than standard visualization techniques such as t-SNE. We experimentally show that our framework improves generalization performance to unseen categories in zero-shot learning. We also provide a finite sample error upper bound guarantee for the method.

## 1 Introduction

Dimensionality reduction is an important problem in machine learning tasks to increase classification performance of learned models, improve computational efficiency, or perform visualization. In the context of visualization, high-dimensional representations are typically converted to two or three-dimensional representations so that the underlying relations between data points can be observed and interpreted from a scatterplot. Currently, a major source of high-dimensional representations that machine learning practitioners have trouble understanding are those generated by deep neural networks. Techniques such as PCA or t-SNE (Van der Maaten, 2014) are typically used to visualize them, *e.g.*, in Law et al. (2017); Snell et al. (2017). Moreover, Schulz et al. (2015) proposed a visualization technique that represents examples based on their predicted category only. However, none of these techniques exploit the fact that deep models have soft probabilistic interpretations. For instance, the output of deep classifiers typically employs softmax regression, which optimizes classification scores across categories by minimizing cross entropy. This results in soft probabilistic representations that reflect the confidence of the model in assigning examples to the different categories. Many other deep learning tasks such as semantic segmentation (Long et al., 2015) or boundary/skeleton detection (Xie & Tu, 2015) also optimize for probability distributions. In this paper, we experimentally demonstrate that the soft probability representations learned by a neural network reveal key structure about the learned model. To this end, we propose a dimensionality reduction framework that transforms probability representations into a low-dimensional space for easy visualization.

Furthermore, our approach improves generalization. In the context of zero-shot learning where novel categories are added at test time, deep learning approaches often learn high-dimensional representations that over-fit to training categories. By learning low-dimensional representations that match the classification scores of a high-dimensional pre-trained model, our approach takes into account inter-class similarities and generalizes better to unseen categories than standard approaches.

**Proposed approach:** We propose to exploit as input representations the probability scores generated by a high-dimensional pre-trained model, called the *teacher* model or *target*, in order to train a

lower-dimensional representation, called the *student*. In detail, our approach learns low-dimensional student representations of examples such that, when applying a specific soft clustering algorithm on the student representations, the predicted clustering scores are similar to the target probability scores.

**Contributions:** This paper makes the following contributions: (1) We propose the first dimensionality reduction approach optimized to consider some soft target probabilistic representations as input. (2) By exploiting the probability representations generated by a pre-trained model, our approach reflects the learned semantic structure better than standard visualization approaches. (3) We experimentally show that our approach improves generalization performance in zero-shot learning. (4) We theoretically analyze the statistical properties of the approach and provide a finite sample error upper bound guarantee for it.

## 2 DIMENSIONALITY REDUCTION FOR PROBABILISTIC REPRESENTATIONS (DRPR)

Our method, called *Dimensionality Reduction of Probabilistic Representations* (DRPR, pronounced *"Derper"*), is given probability distribution representations generated from high-dimensional data as target. Its goal is to learn a low-dimensional representation such that the soft clustering scores predicted by a soft clustering algorithm are similar to the target. If the targets are probability distributions generated by a pre-trained classifier, we want the low-dimensional space to reflect the relationships between categories interpreted by the classifier. The position of each example in the low-dimensional space should then reflect the ambiguity of the classifier for the example (see Fig. 1). We summarize in Section 2.1 the soft clustering algorithm that is used by DRPR in the low-dimensional space, the algorithm is detailed in Banerjee et al. (2005). The general learning algorithm of DRPR is introduced in Section 2.2.

### 2.1 FAMILY OF EFFICIENT SOFT CLUSTERING ALGORITHMS

**Probability density:** We consider that we are given a set of $n$ vectors $\mathbf{f}_1, \cdots, \mathbf{f}_n \in \mathcal{V}$ concatenated into a single matrix $F = [\mathbf{f}_1, \cdots, \mathbf{f}_n]^\top \in \mathcal{V}^n$. In the following, we consider $\mathcal{V} = \mathbb{R}^d$, and $\mathcal{V}^n = \mathbb{R}^{n \times d}$. The goal is to partition $n$ examples into $k$ soft clusters. Each cluster $\mathscr{C}_c$ with $c \in \{1, \cdots, k\}$ has a center $\hat{\boldsymbol{\mu}}_c \in \mathcal{V}$ and its corresponding probability density is $p_c(\mathbf{f}_i) = \exp(-\mathsf{d}(\mathbf{f}_i, \hat{\boldsymbol{\mu}}_c)) \, \mathsf{b}(\mathbf{f}_i)$, where $\mathsf{d}$ is a *regular Bregman divergence* (Banerjee et al., 2005; Bregman, 1967) and $\mathsf{b} : \mathcal{V} \to \mathbb{R}_+$ is a uniquely determined function that depends on $\mathsf{d}$ and ensures that the integral of the density over $\mathcal{V}$ is 1 (*e.g.*, $\mathsf{b}(\mathbf{f}_i) = \sqrt{1/(2\pi)^d}$ if $\mathsf{d}$ is the squared Euclidean distance). For simplicity, we consider that $\mathsf{d}$ is the squared Euclidean distance. The density $p_c(\mathbf{f}_i)$ decreases as the divergence between the example $\mathbf{f}_i$ and the center $\hat{\boldsymbol{\mu}}_c$ increases.

**Bregman Soft Clustering problem (BSCP):** The BSCP (Banerjee et al., 2005) is defined as that of learning the maximum likelihood parameters $\Gamma = \{\hat{\boldsymbol{\mu}}_c, \hat{\pi}_c\}_{c=1}^k$ of a mixture model $p(\mathbf{f}_i|\Gamma) = \sum_{c=1}^k \hat{\pi}_c \exp(-\mathsf{d}(\mathbf{f}_i, \hat{\boldsymbol{\mu}}_c)) \, \mathsf{b}(\mathbf{f}_i)$ where $\hat{\pi}_c$ is the prior probability that $\mathbf{f}_i$ is generated by $\mathscr{C}_c$. To partition the $n$ examples into $k$ clusters, Banerjee et al. (2005) apply the EM algorithm to maximize the likelihood parameter estimation problem for mixture models formulated as: $\max_\Gamma \, \prod_{i=1}^n p(\mathbf{f}_i|\Gamma)$

**Assignment matrix:** Partitioning the $n$ observations in $F$ into $k$ soft clusters is equivalent to determining some soft assignment matrix $\hat{Y} \in (0, 1)^{n \times k}$ in the set $\mathbb{Y}^{n \times k}$ of matrices whose rows are positive and sum to 1. Formally, $\mathbb{Y}^{n \times k}$ is written $\mathbb{Y}^{n \times k} := \{\hat{Y} \in (0, 1)^{n \times k} : \hat{Y}\mathbf{1}_k = \mathbf{1}_n\}$ where $\mathbf{1}_k \in \{1\}^k$ is the $k$-dimensional vector containing only 1. For a given value of $\Gamma$, the element $\hat{Y}_{ic} = p(\mathscr{C}_c|\mathbf{f}_i) \in (0, 1)$ is the posterior probability, or *responsibility* of $\mathscr{C}_c$ for $\mathbf{f}_i$. The higher the value of $\hat{Y}_{ic}$, the more likely $\mathbf{f}_i$ belongs to cluster $\mathscr{C}_c$.

**Local maximum condition:** Once the BSCP has converged to a local maximum of $\max_\Gamma \, \prod_{i=1}^n p(\mathbf{f}_i|\Gamma)$, the following equations are all satisfied:

$$\text{(E-step)} \quad \forall i, c, \;\; p(\mathscr{C}_c|\mathbf{f}_i) = \hat{Y}_{ic} = \frac{\hat{\pi}_c \exp(-\mathsf{d}(\mathbf{f}_i, \hat{\boldsymbol{\mu}}_c))}{\sum_{m=1}^k \hat{\pi}_m \exp(-\mathsf{d}(\mathbf{f}_i, \hat{\boldsymbol{\mu}}_m))} \tag{1}$$

$$\text{(M-step)} \quad \forall c, \; \hat{\boldsymbol{\mu}}_c = \frac{\sum_{i=1}^n \hat{Y}_{ic}\mathbf{f}_i}{\sum_{i=1}^n \hat{Y}_{ic}} \quad \text{and} \quad \forall c, \; \hat{\pi}_c = \frac{1}{n}\sum_{i=1}^n \hat{Y}_{ic} \tag{2}$$

Eq. (1) corresponds to the E-step of the EM algorithm that computes $p(\mathscr{C}_c|\mathbf{f}_i) = \hat{Y}_{ic}$ when the parameters $F$ and $\Gamma$ are given. Eq. (2) corresponds to the M-step, which has a simple form since the likelihood is a regular exponential family function (Banerjee et al., 2005). The M-step may be computationally expensive for other types of exponential family distributions (Banerjee et al., 2005, Section 5.2). It is worth noting that these optimality conditions do not depend on the function b used to define $p_c(\mathbf{f}_i)$, so b can be ignored.

## 2.2 LEARNING THE MAPPING TO THE LOW-DIMENSIONAL SPACE

Section 2.1 explains how to perform soft clustering on some fixed representation $F$. We now describe how to learn $F$ so that the soft clustering scores predicted by the BSCP match those of the target. We assume that we are given the probability representation $\mathbf{y}_i \in [0, 1]^k$ as target for each training example $\mathbf{f}_i$. These representations are concatenated into a single matrix $Y = [\mathbf{y}_1, \cdots, \mathbf{y}_n]^\top \in \mathbb{Y}^{n \times k}$ which is the target of our method for $F$. DRPR learns the representation of $F \in \mathcal{V}^n$ such that the soft assignment matrix obtained from applying the BSCP on $F$ is close to $Y$. We first give the formulation of our prediction function in Eq. (3). Our dimensionality reduction problem is given in Eq. (4).

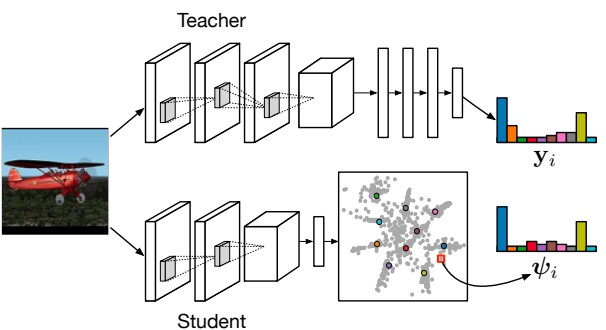

Figure 1: DRPR learns low-dimensional representations that reflect the uncertainties of a pre-trained classifier (here for categories *blue* and *yellow* represented by their centers)

**Prediction function:** Let us assume that we are given the dataset matrix $F = [\mathbf{f}_1, \cdots, \mathbf{f}_n]^\top \in \mathcal{V}^n = \mathbb{R}^{n \times d}$, the centers $M = [\boldsymbol{\mu}_1, \cdots, \boldsymbol{\mu}_k]^\top \in \mathcal{V}^k = \mathbb{R}^{k \times d}$ and the priors $\boldsymbol{\pi} = [\pi_1, \cdots, \pi_k]^\top \in (0, 1)^k$. As in Eq. (1), we define our prediction function $\boldsymbol{\psi}(F, M, \boldsymbol{\pi}) = \Psi \in \mathbb{Y}^{n \times k}$ as the soft assignment matrix predicted by the BSCP given $F$, $M$ and $\boldsymbol{\pi}$. The elements of the matrix $\Psi$ are then computed as follows:

$$\forall i, c, \quad \Psi_{ic} = \frac{\pi_c \exp(-\mathsf{d}(\mathbf{f}_i, \boldsymbol{\mu}_c))}{\sum_{m=1}^k \pi_m \exp(-\mathsf{d}(\mathbf{f}_i, \boldsymbol{\mu}_m))} \tag{3}$$

**Optimization problem:** DRPR learns the representation of $F$ so that the predicted assignment matrix $\boldsymbol{\psi}(F, M, \boldsymbol{\pi}) = \Psi$ is similar to $Y$. Given the optimal condition properties of the BSCP stated in Section 2.1, the optimal values of $M$ and $\boldsymbol{\pi}$ also depend on $\Psi$ and are therefore variables of our dimensionality reduction problem that we formulate:

$$\min_{F, M, \boldsymbol{\pi}} \Delta_n \left( \boldsymbol{\psi}(F, M, \boldsymbol{\pi}), Y \right) \tag{4}$$

The function $\Delta_n(\Psi, Y)$ is an empirical discrepancy loss between the predicted assignment matrix $\Psi$ and the target assignment matrix $Y$. Since the rows of $\Psi$ and $Y$ represent probability distributions, we formulate $\Delta_n$ as the average $\mathsf{KL}$-divergence between the rows of $Y$ and the rows of $\Psi$. Let $\boldsymbol{\psi}_i^\top$ and $\mathbf{y}_i^\top$ be the $i$-th rows of $\Psi$ and $Y$, respectively, we formulate $\Delta_n(\Psi, Y) = \frac{1}{n} \sum_{i=1}^n D_{\mathsf{KL}}(\mathbf{y}_i \| \boldsymbol{\psi}_i)$. Note that the choice of the discrepancy loss $\Delta_n$ is independent of the chosen Bregman divergence $\mathsf{d}$. Moreover, the number of classes $k$ has an impact on the number of clusters in the low-dimensional space but is not related to the dimensionality $d$ of the model. DRPR considers that each class $c \in \{1, \cdots, k\}$ is represented by one cluster prototype $\boldsymbol{\mu}_c \in \mathbb{R}^d$.

**Terminology:** In our experiments, the target (or teacher) is the assignment matrix $Y \in \mathbb{Y}^{n \times k}$ that contains the probability scores generated by a pre-trained neural network. It corresponds to the output of a classifier trained with softmax regression in the visualization experiments, and to the matrices $Y_1$ and $Y_2$ described in Section 4.2. The goal is then to learn $F$, $M$ and $\boldsymbol{\pi}$ so that (1) $F$, $M$ and $\boldsymbol{\pi}$ reach the BSCP optimality conditions given in Section 2.1 and (2) $\Psi = \boldsymbol{\psi}(F, M, \boldsymbol{\pi})$ is similar to $Y$.

**Visualization:** DRPR can be used for visualization since many models (*e.g.*, usual neural networks) have probabilistic interpretations *w.r.t.* the $k$ training categories. In our visualization task, the matrices $M$ and $\boldsymbol{\pi}$ are not provided, whereas the target matrix $Y$ is given. By using the optimality conditions

---

**Algorithm 1** Dimensionality Reduction of Probabilistic Representations (DRPR)

---

**input** : Set of training examples (*e.g.*, images) in $\mathcal{X}$ and their target probability scores (*e.g.*, classification scores *w.r.t.* $k$ training categories), nonlinear mapping $g_\theta$ parameterized by parameters $\theta$, number of iterations $t$
 1: **for** iteration 1 to $t$ **do**
 2:    Randomly sample $n$ training examples $\mathbf{x}_1, \cdots, \mathbf{x}_n \in \mathcal{X}$ and create target assignment matrix $Y \in \mathbb{Y}^{n \times k}$ containing the target probability scores $\mathbf{y}_1, \cdots, \mathbf{y}_n$ (*i.e.*, $Y = [\mathbf{y}_1, \cdots, \mathbf{y}_n]^\top \in \mathbb{Y}^{n \times k}$)
 3:    Create matrix $F \leftarrow [\mathbf{f}_1, \cdots, \mathbf{f}_n]^\top \in \mathcal{V}^n$ such that $\forall i, \mathbf{f}_i = g_\theta(\mathbf{x}_i)$
 4:    Create matrix of centers $M \leftarrow \mathrm{diag}(Y^\top \mathbf{1}_n)^{-1} Y^\top F$ and prior vector $\boldsymbol{\pi} \leftarrow \frac{1}{n} Y^\top \mathbf{1}_n$
 5:    Update the parameters $\theta$ by performing a gradient descent iteration of $\Delta_n(\boldsymbol{\psi}(F, M, \boldsymbol{\pi}), Y)$ (*i.e.*, Eq. (4))
 6: **end for**
**output** : nonlinear mapping $g_\theta$

---

in Eq. (2), we can write the desired values of $M$ and $\boldsymbol{\pi}$ as a function of $F$ and $Y$: at each iteration, for some current value of $F$, the optimal values $M = \mathrm{diag}(Y^\top \mathbf{1}_n)^{-1} Y^\top F$ and $\boldsymbol{\pi} = \frac{1}{n} Y^\top \mathbf{1}_n$ are computed and $F$ is updated via gradient descent. DRPR is illustrated in Algorithm 1 in the case where $F$ is the output of a model $g_\theta$ parameterized by $\theta$, *e.g.*, $g_\theta$ can be a neural network. However, we represent $F$ as non-parametric embeddings in our visualization experiments to have an equitable comparison to non-parametric baselines. The learning algorithm then modifies the matrix $F$ at each iteration. If the priors $\pi_c$ are all equal, then the priors are updated in step 4 as follows: $\boldsymbol{\pi} \leftarrow \frac{1}{k} \mathbf{1}_k$.

**Zero-shot learning:** DRPR can be used to improve zero-shot learning generalization since high-dimensional models may overfit to training categories and the goal of zero-shot learning is to generalize to novel categories. In the considered zero-shot learning task, the variable $F$ concatenates image representations (outputs of a neural network) in the same way as step 3 of Algorithm 1, and the variable $M$ concatenates category representations extracted from text (outputs of another neural network). Both $F$ and $M$ are of different nature and are therefore computed as concatenating the outputs of two distinct neural networks taking different sources as input. To optimize Eq. (4), both neural networks are trained jointly by fixing the other neural network during backpropagation.

**Choice of divergence:** In our experiments, we consider the squared Euclidean distance $\mathrm{d}(\mathbf{f}_i, \hat{\boldsymbol{\mu}}_c) = \|\mathbf{f}_i - \hat{\boldsymbol{\mu}}_c\|_2^2$. However, DRPR can be used with any regular Bregman divergence (Banerjee et al., 2005). The algorithm is then identical, with the exception of the chosen divergence d to compute the prediction in Eq. (3).

**Convergence and scalability:** Although our problem has 3 variables ($F$, $M$ and $\boldsymbol{\pi}$), we use the optimal properties of the BSCP to write them as a function of each other (see step 4 of Algo 1). Eq. (4) is then an optimization problem *w.r.t.* only one variable $F$. Since the problem is differentiable wrt $F$ and unconstrained, it is easy to optimize by gradient descent (*e.g.*, back-propagation when training neural networks). Moreover, our loss is nonnegative, hence lower-bounded. It is worth noting that we do not apply multiple iterations of the EM algorithm at each gradient descent iteration, as we first use the optimal properties of BSCP to obtain closed-form formulations of $M$ and $\boldsymbol{\pi}$, and then compute $\boldsymbol{\psi}(F, M, \boldsymbol{\pi}) = \Psi$ to minimize our problem in Eq. (4). Unlike t-SNE and many iterative DR problems, the complexity of DRPR is linear in $n$ (instead of quadratic) and linear in $k$, which makes it efficient and scalable. Our visualization experiments take less than 5 minutes to do $10^5$ iterations while t-SNE takes 1 hour to do 1000 iterations. PCA, which has an efficient closed-form solution, is still much faster. Our approach is thus simple to optimize, hence scalable, and it generalizes to a large family of Bregman divergences.

**Gradient interpretation:** We now interpret the gradient of our optimization problem *w.r.t.* examples. To simplify its formulation, we consider that all the priors $\pi_1, \cdots, \pi_k$ are equal and the matrix $M$ does not depend on $F$, which is the case in our zero-shot learning task.

When d is the squared Euclidean distance, all the priors are equal and $M$ does not depend on $F$, the gradient of Eq. (4) *w.r.t.* $\mathbf{f}_i$ is:

$$\frac{2}{n} \sum_{c=1}^{k} Y_{ic} \Psi_{ic} [\sum_{m=1}^{k} \exp(\|\mathbf{f}_i - \boldsymbol{\mu}_c\|_2^2 - \|\mathbf{f}_i - \boldsymbol{\mu}_m\|_2^2)] [-\boldsymbol{\mu}_c + \sum_{m=1}^{k} \Psi_{im} \boldsymbol{\mu}_m] \qquad (5)$$

One can observe that its magnitude depends on both the target scores $Y_{ic} \in (0, 1)$ and the predicted responsibilities $\Psi_{ic}$. The gradient tries to make the vector $\mathbf{f}_i$ closer to each centroid $\boldsymbol{\mu}_c$ while separating it from all the centroids $\boldsymbol{\mu}_m$ depending on their predicted scores $\Psi_{im} \in (0, 1)$.

**Statistical guarantee:** We analyze the statistical property of the algorithm in Appendix A. Theorem 1 provides a finite sample upper bound guarantee on the quality of the minimizer of the empirical discrepancy of Eq. (4). We show that it is upper bounded by the minimum of the true expected discrepancy, and an estimation error term of $O(n^{-1/2})$ (under certain conditions and with high probability). We defer the detail to the appendix.

## 3 RELATED WORK

This paper introduces a dimensionality reduction method that represents the relations in a dataset that has probabilistic interpretation. It can be seen as a metric learning approach.

**Metric Learning:** The most related approaches try to solve the "supervised" hard clustering problem (Law et al., 2016) . During training, they are given the target hard assignment matrix $Y \in \{0, 1\}^{n \times k}$ where $Y_{ic} = 1$ if the training example $\mathbf{f}_i$ has to belong to $\mathscr{C}_c$, and 0 otherwise. The goal is to learn a representation such that applying a hard clustering algorithm (*e.g.*, kmeans) on the training dataset will return the desired assignment matrix $Y$. These approaches can be decomposed in 2 groups: (1) the methods that optimize some regression problem (Lajugie et al., 2014; Law et al., 2016; 2017) and exploit orthogonal projection properties that hold only in the hard clustering context, and (2) the methods that use exponential families (Mensink et al., 2012; Snell et al., 2017) to describe some probability score. In the latter case, the learning problem is written as a multi-class logistic regression problem where the probability of a category $\mathscr{C}_c$ given an example $\mathbf{f}_i$ is $p(\mathscr{C}_c|\mathbf{f}_i) = \frac{\exp(-\delta_W(\mathbf{f}_i, \boldsymbol{\mu}_c))}{\sum_{e=1}^{k} \exp(-\delta_W(\mathbf{f}_i, \boldsymbol{\mu}_e))}$ and $\delta_W$ is the learned dissimilarity function. DRPR generalizes those approaches and can also be used for hard clustering learning.[1] For instance, Snell et al. (2017) is a special case of DRPR where $\boldsymbol{\pi} = \frac{1}{k}\mathbf{1}_k$, $Y \in \{0, 1\}^{n \times k}$ is a hard assignment matrix and $\Delta_n$ is the same $\Delta_n$ as ours (*i.e.*, $\Delta_n(\Psi, Y) = \frac{1}{n}\sum_{i=1}^{n} D_{\mathsf{KL}}(\mathbf{y}_i \| \boldsymbol{\psi}_i)$ by using the convention $0 \log 0 = 0$). Moreover, the optimal value of $M$ is not implicitly written as a function of $F$ and $Y$ in Snell et al. (2017). The approach in Mensink et al. (2012) is similar to Snell et al. (2017) but only considers linear models. In summary, both Snell et al. (2017) and Mensink et al. (2012) do not exploit the BSCP formulation to its full potential as they consider some restricted hard clustering case. DRPR generalizes these approaches to the soft clustering context at no additional algorithmic complexity.

**Dimensionality reduction:** Learning models by exploiting soft probability scores predicted by another pre-trained model as supervision was proposed in Ba & Caruana (2014) for classification. It was experimentally observed in Ba & Caruana (2014) that using the output of a large pre-trained neural network as supervision, instead of ground truth labels, improves classification performance of small neural networks. However, in Ba & Caruana (2014), each dimension of the student representation describes the confidence score of the model for one training category, which is problematic in contexts such as zero-shot learning where categories can be added or removed at test time. Our approach can learn a representation with dimensionality different from the number of training categories. It can therefore be used in zero-shot learning. Dimensionality reduction with neural networks has been proposed in unsupervised contexts (*e.g.*, to maximize variance in the latent space (Hinton & Salakhutdinov, 2006)) and in supervised contexts (*e.g.*, using ground truth labels as supervision (Salakhutdinov & Hinton, 2007)). Instead, our approach exploits probability scores generated by a teacher pre-trained model. These scores may be very different from ground truth labels if the pre-trained model does not generalize well on the dataset. Our approach can then help understand what was learned by the teacher by visualizing groups of examples the teacher has trouble distinguishing.

---

[1]We experimentally show in the supplementary material that DRPR can be learned for hard supervised clustering when the centers are implicit (see Section C.3).

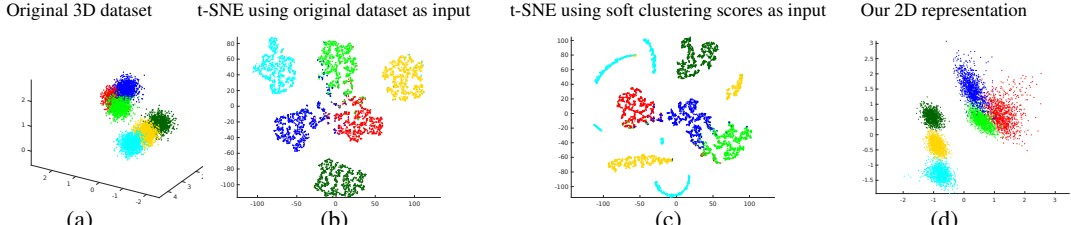

Figure 2: (a) Original 3-dimensional dataset containing 6 clusters (one color per cluster); (b) 2D representation obtained with t-SNE by exploiting the original 3D representation as input; (c) 2D representation obtained with t-SNE by exploiting soft probability scores *w.r.t.* the 6 clusters; (d) 2D representation obtained by our method by exploiting using the same supervision as (c). The relative inter-cluster distances of the original dataset are preserved with our approach, unlike t-SNE.

## 4    EXPERIMENTS

We evaluate the relevance of our method in two types of experiments. The first learns low-dimensional representations for visualization to better interpret pre-trained deep models. The second experiment exploits the probability scores generated by a pre-trained classifier in the zero-shot learning context; these probability scores are used as supervision to improve performance on novel categories.

### 4.1    VISUALIZATION

Interpreting deep models is a difficult task, and one of the most common tools to solve that task is visualization. Representations are most often visualized with t-SNE which does not account for the probabilistic interpretation of the learned models. We propose to exploit probability classification scores as input of our dimensionality reduction framework. In the visualization experiments of this section, DRPR learns non-parametric low-dimensional embeddings (*i.e.* our representations are not outputs of neural networks but vectors) as done by non-parametric baselines. Nonetheless, DRPR can also be learned with neural networks (*e.g.*, as done in Section 4.2).

**Toy dataset:** As an illustrative toy experiment, we compare the behavior of t-SNE and DRPR when applied to reduce a simple artificial 3-dimensional dataset as 2D representations. The 3D dataset illustrated in Fig. 2 (a) contains $k = 6$ clusters, each generated by a Gaussian model and containing 1,000 artificial points. To generate target soft clustering probability scores in the 3D space, we compute the relative distances of the examples to the different cluster centers and normalize them to obtain (probability) responsibilities as done in Eq. (3). In detail, let us note the original 3-dimensional dataset $X = [\mathbf{x}_1, \cdots, \mathbf{x}_n]^\top \in \mathbb{R}^{n \times 3}$ (where $n$ is the number of examples) plotted in Fig. 2 (a). The target soft assignment matrix $Y \in \mathbb{Y}^{n \times k}$ where $k = 6$ is constructed by computing $Y_{ic} = \frac{\exp(-\|\mathbf{x}_i - \mathbf{c}_c\|^2)}{\sum_{e=1}^{k} \exp(-\|\mathbf{x}_i - \mathbf{c}_e\|^2)}$ where $\mathbf{c}_c \in \mathbb{R}^3$ is the center of the $c$-th Gaussian model (defined by its color) in the original space, and priors are equal. We plot in Fig. 2 the 2D representation of our model when using $Y$ as input/target, and the t-SNE representations obtained when using the original dataset $X$ as input in Fig. 2 (b) and $Y$ as input in Fig. 2 (c). Two observations can be made from Fig. 2: (1) the global structure of the original dataset is better preserved with DRPR than with t-SNE; (2) DRPR satisfies the relative distances between the different clusters better than t-SNE since DRPR tries to preserve the relative responsibilities of the different clusters. We quantitatively evaluate these two observations in the following. It is also worth noting that it is known that distances between clusters obtained with t-SNE may not be meaningful (Wattenberg et al., 2016) as t-SNE preserves local neighborhood instead of global similarities.

In the following (*i.e.* in Fig. 4 and tables of results), we only consider the case where t-SNE takes as input the *logits* (*i.e.*, classification scores before the softmax operation) instead of probability scores since the latter case returns bad artifacts such as the one in Fig. 2 (c). Other examples of bad artifacts obtained with t-SNE exploiting probability scores are provided in the supplementary material. We also provide in the supplementary material the visualizations obtained by t-SNE when replacing the $\ell_2$-norm in the input space by the KL-divergence and the Jensen-Shannon divergence to compare probabilistic representations that DRPR uses as input. These lead to worse visualizations than Fig 2 (b) in terms of preserving original clusters and their inter-cluster distances.

Table 1: NPR performance (in %) for different values of $\kappa$

| | | $\kappa$ | 1 | 2 | 5 | 10 | 50 | 100 | | $\kappa$ | 1 | 2 | 5 | 10 | 50 | 100 |
|---|---|---|---|---|---|---|---|---|---|---|---|---|---|---|---|---|
| MNIST | | logits | 6.3 | 8.7 | 12.4 | 16.0 | 28.2 | 36.5 | CIFAR10 | logits | 1.7 | 2.8 | 4.8 | 6.9 | 17.0 | 24.7 |
| | dim reduction | PCA | 0 | 0.1 | 1.3 | 2.4 | 8.5 | 14.1 | | PCA | 0.1 | 0.2 | 0.3 | 0.8 | 3.2 | 5.9 |
| | | LLE | 0 | 0.1 | 0.2 | 1.8 | 4.1 | 8.6 | | LLE | 1.3 | 1.6 | 3.4 | 4.2 | 9.8 | 15.6 |
| | | ISOMAP | 0.5 | 1.0 | 2.0 | 3.3 | 10.4 | 16.7 | | ISOMAP | 0.2 | 0.3 | 0.6 | 1.0 | 4.1 | 7.5 |
| | | t-SNE | 5.2 | 7.4 | 9.8 | 12.1 | 20.4 | 27.7 | | t-SNE | 1.9 | 2.7 | 4.1 | 5.6 | 12.8 | 19.5 |
| | | DRPR (Ours) | **7.0** | **9.4** | **13.4** | **16.3** | **24.5** | **30.3** | | DRPR | **8.4** | **10.2** | **13.2** | **15.2** | **21.1** | **26.6** |
| STL | | logits | 5.9 | 8.4 | 11.8 | 14.8 | 25.5 | 32.4 | CIFAR100 | logits | 2.5 | 4.1 | 6.3 | 8.5 | 15.9 | 19.3 |
| | dim reduction | PCA | 0.1 | 0.2 | 0.7 | 1.3 | 4.9 | 8.4 | | PCA | 0 | 0.1 | 0.2 | 0.4 | 1.6 | 2.8 |
| | | LLE | 0.4 | 0.8 | 1.6 | 2.6 | 6.9 | 11.3 | | LLE | 0 | 0 | 0 | 0.1 | 0.2 | 1.4 |
| | | ISOMAP | 0.1 | 0.4 | 1.0 | 1.7 | 6.1 | 10.0 | | ISOMAP | 0 | 0.1 | 0.4 | 0.6 | 2.3 | 3.9 |
| | | t-SNE | 5.9 | 8.0 | 10.6 | 12.1 | 19.6 | 26.7 | | t-SNE | 2.4 | 3.7 | 5.0 | 6.5 | 13.4 | 16.1 |
| | | DRPR (Ours) | **13.9** | **18.9** | **25.9** | **31.6** | **45.3** | **52.0** | | DRPR | **9.1** | **12.6** | **17.9** | **24.0** | **52.7** | **58.6** |

**Quantitative evaluation metrics:** We quantify the relevance of our approach with the two following evaluation metrics: (1) an adaptation of the *Neighborhood Preservation Ratio* (NPR) (Van der Maaten & Hinton, 2012): for each image $i$, it counts the ratio of $\kappa$ nearest neighbors of $i$ (*i.e.* that have the closest probability scores *w.r.t.* the KL divergence) that are also in the set of $\kappa$ nearest neighbors in the learned low-dimensional space (*w.r.t.* the Euclidean distance). It is averaged over all images $i$. This metric evaluates how much images that have similar probability scores are close to each other with the student representation. (2) *Clustering Distance Preservation Ratio* (CDPR): we randomly sample $10^5$ triplets of images $(i, i_+, i_-)$ such that the 3 images all belong to different categories and $i$ has closer probability score to $i_+$ than to $i_-$ *w.r.t.* the KL divergence. The metric counts the percentage of times that the learned representation of $i$ is closer to $i_+$ than to $i_-$ *w.r.t.* the Euclidean distance in the low-dimensional representation. This evaluates how well inter-cluster distances are preserved.

Table 2: CDPR performance (in %)

| | model | MNIST | STL | CIFAR10 | CIFAR100 |
|---|---|---|---|---|---|
| | logits | 81.5 | 80.6 | 79.9 | 67.9 |
| dim reduction | PCA | 69.1 | 66.9 | 67.4 | 56.4 |
| | LLE | 53.9 | 55.3 | 56.5 | 53.8 |
| | ISOMAP | 67.9 | 70.1 | 69.5 | 58.7 |
| | t-SNE | 52.8 | 65.9 | 66.5 | 60.4 |
| | DRPR (ours) | **76.7** | **70.5** | **70.0** | **69.0** |

We evaluate our approach on the test sets of the MNIST (LeCun et al., 1998), STL (Coates et al., 2011), CIFAR 10 and CIFAR 100 (Krizhevsky & Hinton, 2009) datasets with pre-trained models that are publicly available and optimized for cross entropy.[2] The dimensionality of the high-dimensional representations is equal to the number of categories in the respective datasets (*i.e.* 10 except for CIFAR 100 that contains 100 categories). Our goal is to visualize the teacher representations with 2-dimensional representations by using their probability scores as target, not the ground truth labels. Quantitative comparisons with standard visualization techniques such as t-SNE, ISOMAP (Tenenbaum et al., 2000), Locally Linear Embedding (LLE) (Roweis & Saul, 2000) and PCA using the 2 leading dimensions are provided in Tables 1 and 2. We also report the scores obtained with the logit representations which are not dimensionality reduction representations but provide an estimate of the behavior of the original dataset. DRPR outperforms the dimensionality reduction baselines *w.r.t.* both evaluation metrics and is competitive with the logit representation. Examples that have similar probability-based representations are closer with our approach than with other dimensionality reduction baselines. DRPR also better preserves inter-cluster distances. It is worth noting that DRPR exploits as much supervision as the "unsupervised" visualization baselines. Indeed, all the compared methods use as input the same source of supervision which is included in the (classifier output) representations given as input.

**Qualitative results:** Visualizations of pre-trained models obtained with DRPR and t-SNE are illustrated in Fig. 4 for MNIST and STL. The visualizations for CIFAR 10 and 100 are in the supplementary material. DRPR representations contain spiked groups at the corners to better reflect examples that have high confidence scores for one category. Indeed, an example in a spike at the corner of a figure has a soft assignment score *w.r.t.* its closest center close to 1. This means that the pre-trained model has very high confidence to assign the example to the corresponding category (see

---

[2]We use the pre-trained models available at `https://github.com/aaron-xichen/pytorch-playground`.

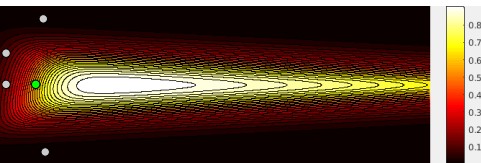

Figure 4: MNIST (left half) and STL (right half) representations of DRPR (left) and t-SNE (right)

illustration in Fig. 3). Examples that are between multiple clusters (usually in the middle of figures) are those that are harder to classify by the model. Detailed visualizations and analysis are provided in the supplementary material.

One can observe that representations obtained with DRPR reflect the semantic structure between categories. On MNIST, categories that contain a curve at the bottom of the digit (*i.e.*, 0, 3, 5, 6, 8 and 9) are in the bottom of Fig. 4 (left); some pairs of digits that are often hard to differentiate by classifiers (*i.e.*, 4 and 9, 1 and 7, 3 and 8) are also adjacent. On STL and CIFAR 10, animal categories are illustrated on the right whereas machines are on the left. Semantically close categories such as *airplane* and *bird*, or *car* and *truck* are also adjacent in the figures. One main difference between the DRPR and t-SNE representations for STL is the distance between the clusters *ship* and *airplane*. These categories are actually hard for the model to differentiate since they contain blue backgrounds and relatively similar objects. In particular, the STL airplane category contains many images of seaplanes lying on the sea and can then be mistaken for ships. This ambiguity between both categories is not observed on the t-SNE representation.

Figure 3: Level sets representing responsibilities of the green cluster whose center is the green circle. Grey circles are the centers of other clusters. The responsibility of the green cluster increases for the examples that are located in the spike at the right side of the figure.

Due to lack of space, a detailed analysis for the CIFAR 100 and STL datasets is available in the supplementary material. A summary of the results is that categories that belong to the same superclass (*e.g.*, categories *hamster, mouse, rabbit, shrew, squirrel* are part of the superclass *small mammals*) are grouped together with DRPR. The DRPR visualization also reflects some semantical structure: plants and insects are on the top left; animals are on the bottom left and categories on the right are outdoor categories. *Medium mammals* are also represented between *small mammals* and *large carnivores*.

In conclusion, the quantitative results show that the representations of DRPR are meaningful since they better preserve the cluster structure and allow observation of ambiguities between categories.

## 4.2 ZERO-SHOT LEARNING

We consider the same zero-shot learning scenario as Reed et al. (2016) and Snell et al. (2017). In particular, we test our approach on the same datasets and splits as them. The main goal is to learn two mappings, one for image representations and one for category representations, in a common space $\mathcal{V}$. The latter mapping takes some category descriptions as input (*e.g.*, from text description or visual attributes). Image representations are then learned so that they are closer to the representative of their category than to the representative of any other category. At test time, categories that were unseen during training are considered, and their representative is obtained by using the second mapping. An image is assigned to the category with closest representative.

**Training datasets:** We use the medium-scaled Caltech-UCSD Birds (CUB) dataset (Welinder et al., 2010) and Oxford Flowers-102 (Flowers) dataset (Nilsback & Zisserman, 2008). CUB contains 11,788 bird images from 200 different species categories split into disjoint sets: 100 categories for training, 50 for validation and 50 for test. Flowers contains 8,189 flower images from 102 different species categories: 62 categories are used for training, 20 for validation and 20 for test. To represent images, Reed et al. (2016) train a GoogLeNet (Szegedy et al., 2015) model whose output dimensionality is 1,024. For each category, Reed et al. (2016) extract some text annotations from

Table 3: Test accuracy on CUB

| Method | Accuracy |
|---|---|
| TMV-HLP Oquab et al. (2014) | 47.9 % |
| SJE Akata et al. (2015) | 50.1% |
| DS-SJE Reed et al. (2016) (Bag-of-words) | 44.1 % |
| DS-SJE Reed et al. (2016) (Char CNN-RNN) | 54.0 % |
| Ziming & Saligrama Zhang & Saligrama (2016) | 55.3 % |
| DS-SJE Reed et al. (2016) (Word CNN-RNN) | 56.8 % |
| Prototypical Networks Snell et al. (2017) | 58.3 % |
| Ours – using DS-SJE (Char CNN-RNN) as supervision | 57.7 % |
| Ours – using Prototypical Networks as supervision | **60.3** % |

Table 4: Test accuracy on Flowers dataset

| Method | Accuracy |
|---|---|
| DS-SJE Reed et al. (2016) (Char CNN) | 47.3 % |
| DS-SJE Reed et al. (2016) (Bag-of-words) | 57.7 % |
| DS-SJE Reed et al. (2016) (Word CNN) | 60.7 % |
| DS-SJE Reed et al. (2016) (Char CNN-RNN) [reported] | 63.7 % |
| DS-SJE Reed et al. (2016) (Char CNN-RNN) [publicly available] | 59.6 % |
| DS-SJE Reed et al. (2016) (Word CNN-RNN) | 65.6 % |
| Prototypical Networks Snell et al. (2017) | 63.9 % |
| Ours – using DS-SJE (Char CNN-RNN) [publicly available] | 62.4 % |
| Ours – using Prototypical Networks as supervision | **68.2** % |

which they learn a representative vector (*e.g.*, based on Char CNN-RNN (Zhang et al., 2015)). The image representations of examples and the text representations of categories are learned jointly so that each image is more similar to the representative vector of its own category than to any other.

**Supervision:** We now describe how we generate the supervision/target of our model from the models pre-trained by (Reed et al., 2016; Snell et al., 2017) and provided by their respective authors.

Once its training is over, Prototypical Network (Snell et al., 2017) represents each image $i$ by some vector $\tilde{\mathbf{f}}_i$ and each category $c$ by some vector $\tilde{\boldsymbol{\mu}}_c$. By concatenating the different vectors into matrices $\tilde{F} = [\tilde{\mathbf{f}}_1, \cdots, \tilde{\mathbf{f}}_n]^\top$ and $\tilde{M} = [\tilde{\boldsymbol{\mu}}_1, \cdots, \tilde{\boldsymbol{\mu}}_k]^\top$ and formulating $\tilde{\boldsymbol{\pi}} = \frac{1}{k}\mathbf{1}_k$, DRPR considers the soft assignment matrix $Y_1 = \boldsymbol{\psi}(\tilde{F}, \tilde{M}, \tilde{\boldsymbol{\pi}}) \in \mathbb{Y}^{n \times k}$ as target (*i.e.* supervision).

In the case of Reed et al. (2016), we consider the same preprocessing as Snell et al. (2017). Each image $i$ is represented by some vector $\check{\mathbf{f}}_i$, each category $c$ is represented by some vector $\check{\boldsymbol{\mu}}_c$ (that is $\ell_2$-normalized in Snell et al. 2017). The target soft assignment matrix of DRPR is then $Y_2 = \boldsymbol{\psi}(\check{F}, \check{M}, \frac{1}{k}\mathbf{1}_k)$ in this case.

**Trained models:** In the model that Snell et al. (2017) provide and that obtains $58.3\%$ accuracy on CUB, ProtoNet trains two models that take as arguments the representations learned by Reed et al. (2016). They train one model $\tilde{g}_{\tilde{\theta}_1}$ for images such that $\forall i, \tilde{\mathbf{f}}_i = \tilde{g}_{\tilde{\theta}_1}(\check{\mathbf{f}}_i)$, and one model $\tilde{g}_{\tilde{\theta}_2}$ for text representative vectors such that $\forall c, \tilde{\boldsymbol{\mu}}_c = \tilde{g}_{\tilde{\theta}_2}(\check{\boldsymbol{\mu}}_c)$. Following Snell et al. (2017), we train two (neural network) models: $g_{\theta_1}$ for images, and $g_{\theta_2}$ for categories. Both of them take as input the image and category representations used to create the target soft assignment matrix (*i.e.*, we take the representations learned by Reed et al. (2016) when its probability scores $Y_2$ are used as supervision, and the representations learned by Snell et al. (2017) otherwise). In this context, we alternately optimize $g_{\theta_1}$ by fixing $M$ (which depends on $g_{\theta_2}$) and optimize $g_{\theta_2}$ by fixing $F$ (which depends on $g_{\theta_1}$).

**Implementation details:** We consider that the learned models $g_{\theta_1}$ and $g_{\theta_2}$ have the same architecture and are multilayer perceptrons (MLP) with *tanh* activation functions. The number of hidden layers $\lambda \in \{0, 1, 2\}$ and output dimensionality $d$ are hyperparameters cross-validated from the accuracy on the validation set. More details on their architecture can be found in the supplementary material.

**Results:** We report the performance of our approach on the test categories of the CUB and Flowers datasets in Tables 3 and 4, respectively. The performance is measured as the average classification accuracy across all unseen classes. We use DS-SJE (Char CNN-RNN) and Prototypical Networks as supervision for our model because they are the only approaches whose pre-trained models are publicly available. Our approach obtains state-of-the-art results on both CUB and Flowers datasets by significantly improving the classification performance of the different classifiers. For instance, it improves the scores of $63.9\%$ obtained by ProtoNet of Flowers up to $68.2\%$. In general, it improves zero-shot learning performance of the different classifiers by $2\%$ to $4.3\%$.

**Impact of dimensionality:** We report in the supplementary material the performance of our model on both the validation and test sets using different numbers of hidden layers, and ranging the output dimensionality $d$ from 16 to the dimensionality $e$ of the input representations. Except for linear models (*i.e.* $\lambda = 0$), reducing the dimensionality improves generalization. This shows that the zero-shot learning performance of a given model can be significantly improved by taking its prediction scores as supervision of our model. To study the impact of the dimensionality reduction generated DRPR, we also ran the codes of Reed et al. (2016); Snell et al. (2017) by learning representations

with dimensionality smaller than those provided (using the same ranges as those in the tables of the supp. material). This decreased their generalization performance. Therefore, directly learning a low-dimensional representation is not a sufficient condition to generalize well. Our framework that learns representations so that examples with similar ambiguities (*i.e.* similar teacher predictions) are close to each other acts as a semantic regularizer. This is suggested by the fact that test accuracy is improved with DRPR even when $e = d$ (as long as the MLPs contain hidden layers).

It is worth mentioning that one test category of the CUB dataset (Indigo bunting) belongs to the ImageNet dataset (Deng et al., 2009) that was used to pretrain GoogLeNet. By using the train/val/test category splits proposed by Xian et al., we did not observe a change of performance of the different models on CUB.

## 5 CONCLUSION

We have proposed a dimensionality reduction approach such that the soft clustering scores obtained in the low-dimensional space are similar to those given as input. We experimentally show that our approach improves generalization performance in zero-shot learning on challenging datasets. It can also be used to complement t-SNE, as a visualization tool to better understand learned models. In particular, we show that we can give a soft clustering interpretation to models that have probabilistic interpretations. Real-world applications that can be used with DRPR include *distillation*. For instance, when the teacher model is too large to store on a device with small memory (*e.g.*, mobile phone), the student model which has a smaller memory footprint is used instead. Low-dimensional representations can also speed up retrieval tasks. Future work includes applying our approach to the task of distillation in the standard classification task where training categories are also test categories.

## 6 ACKNOWLEDGMENTS

We thank Fartash Faghri and the anonymous reviewers for their helpful comments on early versions of this manuscript. This work was supported by Samsung and the Intelligence Advanced Research Projects Activity (IARPA) via Department of Interior/Interior Business Center (DoI/IBC) contract number D16PC00003. The U.S. Government is authorized to reproduce and distribute reprints for Governmental purposes notwithstanding any copyright annotation thereon. AMF acknowledges funding from the Canada CIFAR AI Chairs Program.

**Disclaimer:** The views and conclusions contained herein are those of the authors and should not be interpreted as necessarily representing the official policies or endorsements, either expressed or implied, of IARPA, DoI/IBC, or the U.S. Government.

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

## A  THEORETICAL ANALYSIS

This section provides a statistical guarantee for the algorithm presented in Section 2.2. We show that the minimizer of the empirical discrepancy (4), which uses a finite number of data points $\mathbf{x}_1, \cdots, \mathbf{x}_n \in \mathcal{X}$, is close to the minimizer of the true expected discrepancy measure, to be defined shortly. Let us define the setup here.

We are given data points $\mathcal{D}_n = \{X_1, \ldots, X_n\}$.[3] We suppose that each $X_i \in \mathcal{X}$ is independent and identically distributed (i.i.d.) with the distribution $\nu \in \mathcal{M}(\mathcal{X})$, where $\mathcal{M}(\mathcal{X})$ is the space of all probability distributions defined over $\mathcal{X}$. The teacher is a fixed function $\phi = [\phi(\cdot; 1), \ldots, \phi(\cdot; k)]$ that maps points in $\mathcal{X}$ to a $k$-dimensional simplex, and provides the target probability distributions. That is, the target $y_i$ for $X_i$ is computed as $y_i = \phi(X_i)$.

Consider a function space $\mathcal{G}$ whose domain is $\mathcal{X}$ and its range is a subset of $\mathbb{R}^d$. This function space might be represented by a DNN, but we do not make such an assumption in our statistical analysis. Given a function $g \in \mathcal{G}$ (called $g_\theta$ in the main article) and the number of clusters $k$, we define $\psi_g(x) = [\psi_g(x; 1), \ldots, \psi_g(x; k)]$ as

$$\psi_g(x; c) = \frac{\pi_c \exp\left(-\|g(x) - \mu_c(g)\|_2^2\right)}{\sum_{b=1}^{k} \pi_b \exp\left(-\|g(x) - \mu_b(g)\|_2^2\right)},$$

with the cluster centres

$$\mu_c(g) = \frac{\mathbb{E}\left[\phi_c(X)g(X)\right]}{\mathbb{E}\left[\phi_c(X)\right]} = \frac{\int \phi_c(x)g(x)\mathrm{d}\nu(x)}{\int \phi_c(x)\mathrm{d}\nu(x)},$$

and the priors

$$\pi_c = \int \phi_c(x)\mathrm{d}\nu(x),$$

for $c \in \{1, \ldots, k\}$ (cf. Section 2.1). Note that $\psi_g(x)$ defines a $k$-dimensional probability distribution, and $\mu_c(g)$ is a mapping of the function $g$ to a point in $\mathbb{R}^d$.

Similarly, given $\mathcal{D}_n$, we define the empirical cluster centres

$$\hat{\mu}_c(g) = \frac{\frac{1}{n}\sum_{i=1}^{n} \phi_c(X_i)g(X_i)}{\frac{1}{n}\sum_{i=1}^{n} \phi_c(X_i)},$$

and their corresponding $\hat{\psi}_g(x) = [\hat{\psi}_g(x; 1), \ldots, \hat{\psi}_g(x; k)]$

$$\hat{\psi}_g(x; c) = \frac{\pi_c \exp\left(-\|g(x) - \hat{\mu}_c(g)\|_2^2\right)}{\sum_{b=1}^{k} \pi_b \exp\left(-\|g(x) - \hat{\mu}_b(g)\|_2^2\right)}.$$

Note that here for simplicity of analysis, we assume that the priors $\pi_c$ are exact, and not estimated from data.

The student's goal is to find a $g$ such that $\psi_g$ is close to $\phi$. The closeness of the student to the teacher is defined based on their KL divergence. Specifically, Algorithm 1 minimizes the distorted empirical discrepancy (4), which can be written as[4]

$$\Delta_n(\hat{\psi}, \phi) = \frac{1}{n}\sum_{i=1}^{n} \mathsf{KL}(\phi(X_i)\|\hat{\psi}(X_i)),$$

where $X_i$s are from dataset $\mathcal{D}_n$. Notice that the distorted empirical discrepancy $\Delta_n(\hat{\psi}, \phi)$ is defined based on $\hat{\psi}$, which uses the empirical centres $\hat{\mu}_c$, instead of the expected centres $\mu_c$. We also define an empirical discrepancy w.r.t. the true $\mu_c$ as $\Delta_n(\psi, \phi)$. This quantity is not accessible to the algorithm.

---

[3]Here we use $X_i$ instead of $\mathbf{x}_i$ in order to emphasize their randomness.

[4]Algorithm 1 does not explicitly specify what function generates $\mathbf{y}$; we need to specify it here, so we use $\phi(x)$ for that purpose.

We evaluate the quality of $g$, and its corresponding $\psi_g$, based on how well, in average, it performs on new points $x \in \mathcal{X}$. We consider the expected $\mathsf{KL}$ divergence between $\psi$ and $\phi$ w.r.t. distribution $\nu$ as the measure of performance. Therefore, the discrepancy is

$$\Delta(\psi, \phi) = \mathbb{E}_{X \sim \nu}\left[\mathsf{KL}(\phi(X)||\psi(X))\right] = \int \mathsf{KL}(\phi(x)||\psi(x))\mathrm{d}\nu(x).$$

The output of Algorithm 1 is the minimizer[5] of $\Delta_n(\hat{\psi}, \phi)$, which we denote by $\hat{g}$

$$\hat{g} \leftarrow \operatorname*{argmin}_{g \in \mathcal{G}} \Delta_n(\hat{\psi}_g, \phi). \tag{6}$$

We also define the minimizer of the discrepancy by $g^*$:

$$g^* \leftarrow \operatorname*{argmin}_{g \in \mathcal{G}} \Delta(\psi_g, \phi).$$

We would like to compare the performance of $\hat{g}$ when evaluated according to discrepancy, that is $\Delta(\psi_{\hat{g}}, \phi)$, and compare it with $\Delta(\psi_{g^*}, \phi)$.

Before stating our results, we enlist our assumptions. We shall remark on them as we introduce.

**Assumption A1 (Samples)** The dataset $\mathcal{D}_n = \{X_i\}_{i=1}^n$ consists of i.i.d. samples drawn from $\nu(\mathcal{X})$.

The i.i.d. assumption simplifies the analysis. With extra effort, one can provide similar results for some classes of dependent processes too. For example, if the dependent process comes from a time series and it gradually "forgets" its past, one may still obtain similar statistical guarantees. Forgetting can be formalized through the notion of "mixing" of the underlying stochastic process (Doukhan, 1994). One can then provide statistical guarantees for learning algorithms under various mixing conditions (Yu, 1994; Meir, 2000; Steinwart & Christmann, 2009; Mohri & Rostamizadeh, 2009; 2010; Farahmand & Szepesvári, 2012).

**Assumption A2 (Bounded Function Space)** Functions in $\mathcal{G}$ are $L$-bounded for some $L > 0$, i.e., $g(x) \in [-L, L]^d$ for any $g \in \mathcal{G}$ and $x \in \mathcal{X}$.

This is a mild and realistic assumption on the function space $\mathcal{G}$, and is mainly here to simplify some steps of the analysis.

**Assumption A3 (Teacher)** Part I) The output of the teacher $\phi$ is a probability distribution, i.e., for any $x \in \mathcal{X}$ and $c = \{1, \ldots, k\}$, we have $\phi(x; c) \geq 0$ and $\sum_{c=1}^k \phi(x; c) = 1$.
Part II) We assume that $\pi_c = \mathbb{E}\left[\phi_c(X)\right]$ is bounded away from zero for all $c$. We set $\pi_{\min} = \min_c \pi_c$.

This first part of the assumption explicitly expresses the fact that the algorithm expects to receive a probability distribution from the teacher. If it does not, for example if $\phi(x; c)$ is negative for some $x$ and $c$ but we treat it as a probability in the calculation of the $\mathsf{KL}$ divergence, the algorithm would not be well-defined.

The second part of this assumption requires that prior probability for each cluster is bounded away from zero, and has the probability at least $\pi_{\min}$. This is a technical assumption used by the proof technique; it might be possible that one can relax this assumption.

We need to make some assumptions about the function space $\mathcal{G}$ and its complexity, i.e., capacity. We use covering number (and its logarithm, i.e., metric entropy) as the characterizer of the complexity. The covering number at resolution $\varepsilon$ is the minimum number of balls with radius $\varepsilon$ required to cover the space $\mathcal{M}$ according to a particular metric. We use $\mathcal{N}(\varepsilon, \mathcal{G}, \|\cdot\|)$ to denote the covering number of $\mathcal{G}$ w.r.t. the norm $\|\cdot\|$, which we shall explicitly specify. As $\varepsilon$ decreases, the covering number increases (or more accurately, the covering number is non-decreasing). For example, the covering number for a $p$-dimensional linear function approximator with constraint on the magnitude of its

---

[5]Since our focus is on the statistical analysis, we ignore the issue of the quality of the optimization procedure, and the fact that one may not find a global minimizer of $\Delta_n(\hat{\psi}_g, \phi)$ within $\mathcal{G}$ for a function space described by a DNN.

functions behaves like $O(\frac{1}{\varepsilon^p})$. A similar result holds when the subgraphs of a function space has a VC-dimension $p$. Function spaces whose covering number grows faster are more complex, and estimating a function within them is more difficult. This leads to larger estimation error. On the other hand, those function spaces are often (but not always) show better function approximation properties too. For more detail and many examples, refer to Györfi et al. (2002); van de Geer (2000); Steinwart & Christmann (2008); Giné & Nickl (2015).

Let us define a norm for the function space $\mathcal{G}$:

$$\|g\|_{\infty,2} = \sup_{x \in \mathcal{X}} \|g(x)\|_2$$

This is a mixed-norm where we compute the $\ell_2$-norm for each $g(x) \in \mathbb{R}^d$, and then take the supremum norm over the resulting $\ell_2$-norm. We use this norm to characterize the covering number of $\mathcal{G}$.

**Assumption A4 (Metric Entropy)** There exists constants $B > 0$ and $0 < \alpha < 1$ such that for any $\varepsilon$, the following covering number (i.e., metric entropy) condition is satisfied:

$$\log \mathcal{N}\left(\varepsilon, \mathcal{G}, \|\cdot\|_{\infty,2}\right) \leq \left(\frac{B}{\varepsilon}\right)^{2\alpha}.$$

The logarithm of the covering number of $\mathcal{G}$ is $O(\frac{1}{\varepsilon^{2\alpha}})$. It grows much faster than the metric entropy of linear models, which is $O(p \log(\frac{1}{\varepsilon}))$. This behaviour is suitable to capture the complexity of large function spaces such as the Sobolev space $\mathbb{W}^k(\mathbb{R}^d)$ and many reproducing kernel Hilbert spaces (RKHS).[6] Note that we use a mixed-norm to define the covering number. The use of supremum norm in the definition might be considered conservative. Using a more relaxed norm, for example based on the empirical $L_p(P_{X_{1:n}})$-norm for some $1 \leq p < \infty$, is an open technical question for future work.

Finally let us define the pointwise loss function

$$l(x; g) = \mathsf{KL}(\phi(x) \| \psi_g(x)) = \sum_{c=1}^{k} \phi(x; c) \log \frac{\phi(x; c)}{\psi_g(x; c)}. \tag{7}$$

Notice that $\Delta_n(\psi_g, \phi) = \frac{1}{n} \sum_{i=1}^{n} l(X_i; g)$ and $\Delta(\psi_g, \phi) = \mathbb{E}_\nu [l(X; g)]$. We define the following function space:

$$\mathcal{L} = \{ x \mapsto l(x, g) \, : \, g \in \mathcal{G} \}. \tag{8}$$

We also define the entropy integral (Dudley integral)

$$\mathcal{J}(\varepsilon) = \int_0^\varepsilon \sqrt{\log 2\mathcal{N}\left(u, \mathcal{G}, \|\cdot\|_{\infty,2}\right)} \, \mathrm{d}u. \tag{9}$$

We are now ready to state the main theoretical result of this section.

**Theorem 1.** *Suppose that Assumptions A1, A2, and A3 hold. Consider $\hat{g}$ obtained by solving* (6).

*There exists a finite $c_1 > 0$ such that for any $\delta > 0$, with probability at least $1 - \delta$, we have*

$$\Delta(\psi_{\hat{g}}, \phi) \leq \min_{g \in \mathcal{G}} \Delta(\psi_g, \phi) + \frac{64\sqrt{2}L\sqrt{d}\, \mathcal{J}\left(\frac{16dL^2 + 2\log_2 k}{16L\sqrt{d}}\right)}{\sqrt{n}} + c_1\left(dL^2 + \log(k)\right)\sqrt{\frac{\log(1/\delta)}{n}} +$$

$$\frac{c_2 L\sqrt{d}}{\pi_{min}}\left[\frac{d\mathcal{J}(2L)}{\sqrt{n}} + L\sqrt{\frac{d\ln(k/\delta)}{n}}\right].$$

*Furthermore, if the metric entropy satisfies Assumption A4, there exist constants $c_2, c_4 > 0$ and a function $c_3(\alpha) > 0$, which depends only on $\alpha$, such that for any $\delta > 0$, with probability at least $1 - \delta$, we have*

$$\Delta(\psi_{\hat{g}}, \phi) \leq \min_{g \in \mathcal{G}} \Delta(\psi_g, \phi) + c_2\left[1 + c_3(\alpha)\left(\frac{BL\sqrt{d}}{dL^2 + \log k}\right)^\alpha\right]\left(dL^2 + \log(k)\right)\sqrt{\frac{\log(1/\delta)}{n}} +$$

$$\frac{c_4}{\pi_{min}}\left[\frac{L^{2-\alpha}B^\alpha d^{3/2}}{1 - \alpha} + L^2 d\right]\sqrt{\frac{\log(k/\delta)}{n}}.$$

---

[6] For $\mathbb{W}^k(\mathbb{R}^d) = \mathbb{W}^{k,2}(\mathbb{R}^d)$, the Sobolev space defined w.r.t. the $L_2$-norm of the weak derivatives, we can set $\alpha = \frac{d}{2k}$, see e.g., Lemma 20.6 of Györfi et al. (2002).

This theorem provides a finite sample error upper bound on the true discrepancy of $\hat{g}$, and relates it to the expressiveness and complexity of the function space $\mathcal{G}$, the number of samples $n$, and some other properties. The term $\min_{g \in \mathcal{G}} \Delta(\psi_g, \phi) = \Delta(\psi_{g^*}, \phi)$ is the function approximation error, and reflects the expressiveness of $\mathcal{G}$. This is the minimum achievable error given the function space $\mathcal{G}$. The other terms in the upper bound correspond to the estimation error caused by having a finite number of data points.

Let us focus on the second part of the theorem, which is under the particular choice of covering number according to Assumption A4. In that case, the estimation error shows $n^{-1/2}$ dependence on the number of samples, and hence decreases as we have more training data. We observe that the upper bound increases as the range $L$ of the function space $\mathcal{G}$, the dimension $d$ of the low-dimensional space, and the number of clusters $k$ increases.

The effect of using the distorted empirical discrepancy $\Delta_n(\hat{\psi}_g, \phi)$ instead of $\Delta_n(\psi_g, \phi)$ shows itself in the last term, i.e., the term with the constant multiplier of $\frac{c_2 L \sqrt{d}}{\pi_{\min}}$ in the first part of the theorem and $\frac{c_4}{\pi_{\min}}$ in the second part. Note the appearance of $\pi_{\min}$ in the denominator of the last term. This causes a large error if one of the clusters is improbable. Also observe that if we have $k$ clusters, $\pi_{\min} \leq \frac{1}{k}$ (equality under the uniform distribution over classes), so we have at least a linear dependence on $k$ in the upper bound. This dependence might be due to our proof technique; it remains to be seen whether this can be improved.

*Proof.* To simplify the notation, we denote $\Delta_n(g) = \Delta_n(\psi_g, \phi)$, $\hat{\Delta}_n(g) = \Delta_n(\hat{\psi}_g, \phi)$, and $\Delta(g) = \Delta(\psi_g, \phi)$. We want to relate $\Delta(\hat{g})$ to $\Delta(g^*)$, the supremum of the empirical process $\Delta(g) - \Delta_n(g)$ and the supremum of the distortion of the empirical loss $\Delta_n(g) - \hat{\Delta}_n(g)$. We have the following relations:

$$\Delta(\hat{g}) = \Delta_n(\hat{g}) + (\Delta(\hat{g}) - \Delta_n(\hat{g}))$$
$$= \hat{\Delta}_n(\hat{g}) + \left(\Delta_n(\hat{g}) - \hat{\Delta}_n(\hat{g})\right) + (\Delta(\hat{g}) - \Delta_n(\hat{g}))$$
$$\leq \hat{\Delta}_n(g^*) + \left(\Delta_n(\hat{g}) - \hat{\Delta}_n(\hat{g})\right) + (\Delta(\hat{g}) - \Delta_n(\hat{g}))$$
$$= \Delta_n(g^*) + \left(\hat{\Delta}_n(g^*) - \Delta_n(g^*)\right) + \left(\Delta_n(\hat{g}) - \hat{\Delta}_n(\hat{g})\right) + (\Delta(\hat{g}) - \Delta_n(\hat{g}))$$
$$= \Delta(g^*) + (\Delta_n(g^*) - \Delta(g^*)) + \left(\hat{\Delta}_n(g^*) - \Delta_n(g^*)\right) + \left(\Delta_n(\hat{g}) - \hat{\Delta}_n(\hat{g})\right) + (\Delta(\hat{g}) - \Delta_n(\hat{g}))$$
$$\leq \Delta(g^*) + 2 \sup_{g \in \mathcal{G}} |\Delta(g) - \Delta_n(g)| + 2 \sup_{g \in \mathcal{G}} \left|\Delta_n(g) - \hat{\Delta}_n(g)\right|. \tag{10}$$

The first inequality is because of the the optimizer property of $\hat{g}$, i.e., $\hat{\Delta}_n(\hat{g}) \leq \hat{\Delta}_n(g)$ for any $g \in \mathcal{G}$, including $g^*$.

We need to upper bound the supremum of the empirical process, that is $\sup_{g \in \mathcal{G}} |\Delta(g) - \Delta_n(g)|$, and the supremum of the distortion caused by using $\hat{\psi}$ in minimizing $\Delta_n$ instead of $\psi$, that is $\sup_{g \in \mathcal{G}} |\Delta_n(g) - \hat{\Delta}_n(g)|$, cf (6).

**Upper Bounding** $\sup_{g \in \mathcal{G}} |\Delta(g) - \Delta_n(g)|$. We use Lemma 7 in Appendix B.3 in order to upper bound the supremum of the empirical process, $\sup_{g \in \mathcal{G}} |\Delta(g) - \Delta_n(g)|$, which is equivalent to $\sup_{l \in \mathcal{L}} \left|\frac{1}{n} \sum_{i=1}^n l(X_i) - \mathbb{E}[l(X)]\right|$. That lemma, which is originally Theorem 2.1 of Bartlett et al. (2005), relates the supremum of the empirical process to the Rademacher complexity of $\mathcal{L}$, defined in the same appendix.

To apply the lemma, we first provide upper bound on $l(x)$ and $\text{Var}[l(X)]$. By assumption, $g(x)$ is within $[-L, L]^d$. It is not difficult to see that $\mu_c(g)$ is also within $[-L, L]^d$; for example, see the proof leading to (19). So

$$\|g(x) - \mu_c(g)\|_2^2 \leq 4dL^2.$$

We evoke Proposition 4 with the choice of $f(x; c) = \|g(x) - \mu_c\|_2^2$ and $L' = 4dL^2$ to obtain that

$$l(x; g) \leq 8dL^2 + \log_2 k. \tag{11}$$

Since $l(x; g)$ is bounded, we have

$$\mathrm{Var}\left[l(X; g)\right] \leq \mathbb{E}\left[l(X; g)^2\right] \leq \left(8dL^2 + \log_2 k\right)^2.$$

By the choice of $\beta = 1$, $B = (8dL^2 + \log_2 k)$, and $r = (8dL^2 + \log_2 k)^2$ in Lemma 7, we get that for any $\delta_1 > 0$,

$$\sup_{l \in \mathcal{L}} \left| \frac{1}{n} \sum_{i=1}^{n} l(X_i) - \mathbb{E}\left[l(X)\right] \right| \leq 4\mathbb{E}\left[R_n(\mathcal{L})\right] + (8dL^2 + \log_2 k)\sqrt{\frac{2\log(2/\delta_1)}{n}} +$$
$$2(8dL^2 + \log_2 k)\frac{5}{6}\frac{\log(2/\delta_1)}{n}, \tag{12}$$

with probability at least $1 - \delta_1$.

It remains to provide an upper bound on $\mathbb{E}\left[R_n(\mathcal{L})\right]$. This can be done by using Dudley's integral to relate the Rademacher complexity of $\mathcal{L}$ to the covering number of $\mathcal{L}$. Afterwards, we use Lemma 3 in Appendix A.1 to relate the covering number of $\mathcal{L}$ to the covering number of $\mathcal{G}$. We have

$$\mathbb{E}\left[R_n(\mathcal{L})\right] \leq \frac{4\sqrt{2}}{\sqrt{n}}\mathbb{E}\left[\int_0^{\mathrm{diam}(\mathcal{L})} \sqrt{\log 2\mathcal{N}\left(u, \mathcal{L}, L_2(P_{X_{1:n}})\right)}\, \mathrm{d}u\right]$$
$$\leq \frac{4\sqrt{2}}{\sqrt{n}}\mathbb{E}\left[\int_0^{16dL^2 + 2\log_2 k} \sqrt{\log 2\mathcal{N}\left(u, \mathcal{L}, \|\cdot\|_\infty\right)}\, \mathrm{d}u\right]$$
$$\leq \frac{4\sqrt{2}}{\sqrt{n}}\mathbb{E}\left[\int_0^{16dL^2 + 2\log_2 k} \sqrt{\log 2\mathcal{N}\left(\frac{u}{16L\sqrt{d}}, \mathcal{G}, \|\cdot\|_{\infty,2}\right)}\, \mathrm{d}u\right]$$
$$= \frac{4\sqrt{2}(8L\sqrt{d})\,\mathcal{J}\left(\frac{16dL^2 + 2\log_2 k}{16L\sqrt{d}}\right)}{\sqrt{n}}. \tag{13}$$

In the second inequality, we benefit from two observations: first, we use $l(x) \leq 8dL^2 + \log_2 k$ for any $l \in \mathcal{L}$ (11) to upper bound $\mathrm{diam}(\mathcal{L})$; second, the covering number w.r.t. $L_2(P_{X_{1:n}})$ can be upper bounded by the covering number w.r.t. the supremum norm.[7]

**Upper Bounding** $\sup_{g \in \mathcal{G}} \left| \Delta_n(g) - \hat{\Delta}_n(g) \right|$. We use Proposition 5 in Appendix A.2, which states that for any $\delta_2 > 0$,

$$\sup_{g \in \mathcal{G}} \left| \Delta_n(g) - \hat{\Delta}_n(g) \right| \leq \frac{c_1 L\sqrt{d}}{\pi_{\min}}\left[\frac{d\mathcal{J}(2L)}{\sqrt{n}} + L\left(\sqrt{\frac{d\ln(k/\delta_2)}{n}} + \frac{\ln(dk/\delta_2)}{n}\right)\right]$$
$$\leq \frac{c_2 L\sqrt{d}}{\pi_{\min}}\left[\frac{d\mathcal{J}(2L)}{\sqrt{n}} + L\sqrt{\frac{d\ln(k/\delta_2)}{n}}\right], \tag{14}$$

with probability at least $1 - \delta_2$.

Plugging (12) and (14) in (10) and using the entropy integral upper bound (13) lead to the desired result of the first part.

To prove the second part of the theorem, we use $\log \mathcal{N}\left(\varepsilon, \mathcal{G}, \|\cdot\|_{\infty,2}\right) \leq \left(\frac{B}{\varepsilon}\right)^{2\alpha}$ to calculate $\mathcal{J}(\varepsilon)$, which results in

$$\mathcal{J}(\varepsilon) \leq \frac{B^\alpha \varepsilon^{1-\alpha}}{1 - \alpha}.$$

By plugging in $\varepsilon = 16dL^2 + 2\log_2 k$, we get that

$$\mathbb{E}\left[R_n(\mathcal{L})\right] \leq \frac{32\sqrt{2}(BL\sqrt{d})^\alpha \left(8dL^2 + \log_2 k\right)^{1-\alpha}}{(1-\alpha)\sqrt{n}}.$$

---

[7]This version of Dudley's integral is from Theorem 2.3.7 of Giné & Nickl (2015). We use it similar to the argument in their proof of Theorem 3.5.1 and the comments thereafter. One could also use Theorem A.7 by Bartlett et al. (2005) (originally from Dudley) and note that the upper bound of the integral does not need to go up to infinity when the function space is bounded in the norm.

We upper bound $\mathcal{J}(2L)$ in (14) to obtain:

$$\mathcal{J}(2L) \leq \frac{2^{1-\alpha} B^\alpha L^{1-\alpha}}{1-\alpha}.$$

After some simplifications these lead to the desired result of the second part. $\qquad\square$

## A.1 SOME TECHNICAL TOOLS

We develop some technical tools required in the proof of Theorem 1. Proposition 2 provides Lipschitz constant for some functions that are used in the proof of Lemma 3 to relate the covering number of the function space $\mathcal{L}$, defined in (8), to that of $\mathcal{G}$. This was a key step of the proof of the theorem. Proposition 4 provides an upper bound on the magnitude of $l(x; f)$, shortly defined in (16).

We introduce a few more notations to reduce the clutter. Let $d_g(x; c) = \|g(x) - \mu_c(g)\|_2^2$, so we can write

$$\psi_g(x; c) = \frac{\pi_c \exp\left(-d_g(x; c)\right)}{\sum_{b=1}^k \pi_b \exp\left(-d_g(x; b)\right)}. \qquad c \in \{1, \ldots, k\}$$

For a function $f : \mathcal{X} \times \{1, \ldots, k\} \to \mathbb{R}$, we define

$$p_f(x; c) = \frac{\pi_c \exp\left(-f(x; c)\right)}{\sum_{b=1}^k \pi_b \exp\left(-f(x; b)\right)}. \qquad c \in \{1, \ldots, k\} \qquad (15)$$

We overload the pointwise loss function $l(x; g)$, defined in (7), and define a similar definition for $l(x; f)$ as follows

$$l(x; f) = \sum_{c=1}^k \phi(x; c) \log \frac{\phi(x; c)}{p_f(x; c)}. \qquad (16)$$

It is clear that with the choice of $f = d_g$, the probability distribution $p_f$ is the same as $\psi_g$.

The following proposition specifies the Lipschitz properties of $d_g$ and $l(x; f)$.

**Proposition 2.** *Suppose that Assumption A3 hold.*

*Part I) Consider $g_1, g_2 \in \mathcal{G}$ and let Assumption A2 hold. For any $x \in \mathcal{X}$ and $c \in \{1, \ldots, k\}$, we have*

$$|d_{g_1}(x; c) - d_{g_2}(x; c)| \leq 4L\sqrt{d}\left[\|g_1(x) - g_2(x)\|_2 + \sup_{x' \in \mathcal{X}} \|g_1(x') - g_2(x')\|_2\right].$$

*Part II) Consider two functions $f_1, f_2 : \mathcal{X} \times \{1, \ldots, k\} \to \mathbb{R}$. For any $x \in \mathcal{X}$ we have*

$$|l(x; f_1) - l(x; f_2)| \leq 2 \|f_1(x; \cdot) - f_2(x; \cdot)\|_\infty.$$

*Proof.* Part I) First notice that for any two vectors $u$ and $v$, by the Cauchy-Schwarz inequality we have

$$\left|\|u\|_2^2 - \|v\|_2^2\right| = \left|\sum_i (u_i^2 - v_i^2)\right| = \left|\sum_i (u_i - v_i)(u_i + v_i)\right|$$

$$\leq \left|\sqrt{\sum_i (u_i - v_i)^2}\sqrt{\sum_i (u_i + v_i)^2}\right| = \|u - v\|_2 \|u + v\|_2.$$

Consider $g_1, g_2 \in \mathcal{G}$, and their corresponding $d_{g_1}$ and $d_{g_2}$. We have

$$|d_{g_1}(x; c) - d_{g_2}(x; c)| = \left|\|g_1(x) - \mu_c(g_1)\|_2^2 \|g_2(x) - \mu_c(g_2)\|_2^2\right|$$

$$\leq \|(g_1(x) - \mu_c(g_1)) - (g_2(x) - \mu_c(g_2))\|_2 \times$$
$$\|(g_1(x) - \mu_c(g_1)) + (g_2(x) - \mu_c(g_2))\|_2$$
$$\leq [\|g_1(x) - g_2(x)\|_2 + \|\mu_c(g_1) - \mu_c(g_2)\|_2] \times$$
$$\|(g_1(x) + g_2(x)) - (\mu_c(g_1) + \mu_c(g_2))\|_2. \qquad (17)$$

Note that

$$\|\mu_c(g_1) - \mu_c(g_2)\|_2 = \frac{\left\|\int \phi_c(x)(g_1(x) - g_2(x))\mathrm{d}\nu(x)\right\|_2}{\int \phi_c(x)\mathrm{d}\nu(x)} \leq \frac{\int \phi_c(x)\|g_1(x) - g_2(x)\|_2\,\mathrm{d}\nu(x)}{\int \phi_c(x)\mathrm{d}\nu(x)}$$

$$\leq \frac{\sup_x \|g_1(x) - g_2(x)\|_2 \int \phi_c(x)\mathrm{d}\nu(x)}{\int \phi_c(x)\mathrm{d}\nu(x)} = \sup_x \|g_1(x) - g_2(x)\|_2, \qquad (18)$$

where we used Jensen's inequality and the fact that $\phi_c(x) \geq 0$. As $\mu_c(0) = 0$, we also obtain that

$$\|\mu_c(g)\|_2 = \|\mu_c(g) - \mu_c(0)\|_2 \leq \sup_x \|g(x)\|_2. \qquad (19)$$

As $g(x) \in [-L, L]^d$, it holds that $\|g(x)\|_2 \leq L\sqrt{d}$. Therefore by (17), (18), and (19),

$$|d_{g_1}(x; c) - d_{g_2}(x; c)| \leq 4L\sqrt{d}\left[\|g_1(x) - g_2(x)\|_2 + \|\mu_c(g_1) - \mu_c(g_2)\|_2\right]$$

$$\leq 4L\sqrt{d}\left[\|g_1(x) - g_2(x)\|_2 + \sup_{x'} \|g_1(x') - g_2(x')\|_2\right].$$

Part II) For functions $f_1, f_2$, using the definition of $l(x; f)$ (16), we get that

$$l(x; f_1) - l(x; f_2) = \sum_c \phi(x; c) \log \frac{p_{f_2}(x; c)}{p_{f_1}(x; c)}. \qquad (20)$$

By substituting the definition (15) and some simplifications, we get

$$\log \frac{P_{f_2}(x; c)}{P_{f_1}(x; c)} = \log\left[\frac{\frac{\pi_c \exp(-f_2(x;c))}{\sum_{b=1}^k \pi_b \exp(-f_2(x;b))}}{\frac{\pi_c \exp(-f_1(x;c))}{\sum_{b=1}^k \pi_b \exp(-f_1(x;b))}}\right]$$

$$= [f_1(x; c) - f_2(x; c)] + [\rho(-f_2(x; \cdot)) - \rho(-f_1(x; \cdot))], \qquad (21)$$

with

$$\rho(u) = \log\left(\sum_b \pi_b \exp(u_b)\right),$$

where $u \in \mathbb{R}^k$.

We study the Lipschitz property of $\rho(u)$ as a function of $u$ in order to upper bound the second term on the right-hand side (RHS).

We take the derivative of $\rho(u)$ w.r.t. the $c$-th component of $u$ to obtain that

$$\frac{\partial \rho(u)}{\partial u_c} = \frac{\pi_c \exp(u_c)}{\sum_b \pi_b \exp(u_b)} \triangleq q_c(u).$$

Notice that $q_c(u)$ is a probability distribution. We denote $(q_1(u), \ldots, q_k(c))$ by $q(u)$.

By Taylor's theorem, for $u, u' \in \mathbb{R}^k$, we have

$$\rho(u') = \rho(u) + \langle u' - u, \nabla_u \rho(\tilde{u}) \rangle$$

$$= \rho(u) + \langle u' - u, q(\tilde{u}) \rangle,$$

for some $\tilde{u} = (1 - \lambda)u + \lambda u'$ with $0 \leq \lambda \leq 1$. By Hölder inequality, for any Hölder conjugate $\frac{1}{r} + \frac{1}{s} = 1$ ($r, s \in [1, \infty]$), we get

$$|\rho(u') - \rho(u)| \leq \|u' - u\|_r \max_{u \leq \tilde{u} \leq u'} \|q(\tilde{u})\|_s,$$

where $\max$ over $u \leq \tilde{u} \leq u'$ should be understood as the maximum over the line segment between $u$ and $u'$.

In particular,

$$|\rho(u') - \rho(u)| \leq \|u' - u\|_\infty \max_{u \leq \tilde{u} \leq u'} \|q(\tilde{u})\|_1 = \|u' - u\|_\infty.$$

Here we used the fact that $q_c(u)$ is a probability distribution and its sum is equal to 1, for any choice of $\tilde{u}$.

We substitute (21) in (20) and use the upper bound $|\rho(-f_2(x;\cdot)) - \rho(-f_1(x;\cdot))| \leq \|f_1(x;\cdot) - f_2(x;\cdot)\|_\infty$, which is just shown, to get that

$$|l(x;f_1) - l(x;f_2)| \leq 2\|f_1(x;\cdot) - f_2(x;\cdot)\|_\infty,$$

as desired. $\qquad\square$

The following lemma relates the covering number of the function space $\mathcal{L}$ to the covering number of the function space $\mathcal{G}$.

**Lemma 3.** *Consider the function space $\mathcal{G}$ and its induced function space $\mathcal{L}$ (8). Let Assumptions A2 and A3 hold. For any $\varepsilon > 0$, we have*

$$\mathcal{N}(\varepsilon, \mathcal{L}, \|\cdot\|_\infty) \leq \mathcal{N}\left(\frac{\varepsilon}{16L\sqrt{d}}, \mathcal{G}, \|\cdot\|_{\infty,2}\right).$$

*Proof.* By choosing $f(x;c) = d_g(x;c) = \|g(x) - \mu_c(g)\|_2^2$ and using both parts of Proposition 2, we get that

$$|l(x;g_1) - l(x;g_2)| \leq 2\|d_{g_1}(x,\cdot) - d_{g_2}(x,\cdot)\|_\infty$$
$$\leq 8L\sqrt{d}\left[\|g_1(x) - g_2(x)\|_2 + \sup_{x'\in\mathcal{X}}\|g_1(x') - g_2(x')\|_2\right].$$

If we have an $\varepsilon$-cover of $\mathcal{G}$ w.r.t. $\|g\|_{\infty,2} = \sup_{x\in\mathcal{X}}\|g(x)\|_2$, it induces a $16L\sqrt{d}\varepsilon$-cover on $\mathcal{L}$ w.r.t. the supremum norm. $\qquad\square$

The following proposition upper bounds the magnitude of $l(x;f)$.

**Proposition 4.** *Suppose that Assumption A3 hold and $|f(x;c)| \leq L'$ for any $x \in \mathcal{X}$ and $c \in \{1,\ldots,k\}$. Consider $p_f$ defined in (15). It holds that*

$$\sum_{c=1}^k \phi(x;c) \log \frac{\phi(x;c)}{p_f(x;c)} \leq 2L' + \log_2 k.$$

*Proof.* For simplicity, we ignore the dependence on $x$. We use the definition of $p_f$ (15) to get

$$\sum_{c=1}^k \phi(c) \log \frac{\phi(c)}{p_f(c)} = \sum_c \phi(c) \log \phi(c) - \sum_c \phi(c) \log \pi_c + \sum_c \phi(c)f(c) +$$
$$\sum_c \phi(c) \log\left(\sum_b \pi_b \exp(-f(b))\right).$$

Let us consider each term on the RHS.

- As $\log\phi(c) \leq 0$, we have $\sum_c \phi(c) \log \phi(c) \leq 0$.

- The priors $(\pi_1,\ldots,\pi_k)$ indeed defines a probability distribution, as each $\pi_c$ is non-negative and $\sum_c \pi_c = \sum_c \int \phi_c(x)d\nu(x) = \int \sum_c \phi_c(x)d\nu(x) = \int 1 \times d\nu(x) = 1$. So $-\sum_c \phi(c) \log \pi_c$ is the entropy of a probability distribution over an alphabet with size $k$, which is at most $\log_2 k$.

- The summation $\sum_c \phi(c)f(c)$ is upper bounded by $L'$ because of the boundedness of $f(c) \leq L'$ and the fact that $\phi$ is a probability distribution and sums to one.

- Consider the term $\sum_c \phi(c) \log\left(\sum_b \pi_b \exp(-f(b))\right)$. By the boundedness of $f(b)$, we have $\sum_b \pi_b \exp(-f(b)) \leq \sum_b \pi_b \exp(+L') \leq \exp(L')$. Therefore, the term is upper bounded by $\sum_c \phi(c) \log(\exp(L')) \leq L'$.

Collecting all these terms leads to the upper bound of $2L' + \log_2 k$. $\qquad\square$

A.2 CONTROLLING THE DISTORTED LOSS

This section provides an upper bound on the distortion of the empirical discrepancy, i.e., $\sup_{g \in \mathcal{G}} |\Delta_n(g) - \hat{\Delta}_n(g)|$.

**Proposition 5.** *Suppose that Assumptions A1, A2, and A3 hold. For any $\delta > 0$, there exists a constant $c_1 > 0$ such that with probability at least $1 - \delta$, we have*

$$\sup_{g \in \mathcal{G}} \left| \Delta_n(g) - \hat{\Delta}_n(g) \right| \leq \frac{c_1 L \sqrt{d}}{\pi_{min}} \left[ \frac{d \mathcal{J}(2L)}{\sqrt{n}} + L \left( \sqrt{\frac{d \ln(k/\delta)}{n}} + \frac{\ln(dk/\delta)}{n} \right) \right].$$

*Proof.* Recall that $d_g(x; c) = \|g(x) - \mu_c(g)\|_2^2$, $\Delta_n(g) = \Delta_n(\psi_g, \phi)$, and $\hat{\Delta}_n(g) = \Delta_n(\hat{\psi}_g, \phi)$.

Let us define $\hat{d}_g(x; c) = \|g(x) - \hat{\mu}_c(g)\|_2^2$. Also from the definition of $l(x, f)$ (16), we can write $\Delta_n(\psi_g, \phi) = \frac{1}{n} \sum_{i=1}^n l(X_i; d_g)$. Similarly we have $\hat{\Delta}_n(g) = \Delta_n(\hat{\psi}_g, \phi) = \frac{1}{n} \sum_{i=1}^n l(X_i; \hat{d}_g)$. Therefore, for any $g \in \mathcal{G}$ and by the application of Proposition 2 (Part II), we have

$$\left| \Delta_n(g) - \hat{\Delta}_n(g) \right| = \left| \frac{1}{n} \sum_{i=1}^n l(X_i, d_g) - l(X_i, \hat{d}_g) \right|$$

$$\leq \frac{2}{n} \sum_{i=1}^n \left\| d_g(X_i; \cdot) - \hat{d}_g(X_i; \cdot) \right\|_\infty, \tag{22}$$

where the supremum norm is taken over the centres $c = \{1, \ldots, k\}$. For any $x \in \mathcal{X}$ and any $c = \{1, \ldots, k\}$, we have

$$\left| d_g(x; c) - \hat{d}_g(x; c) \right| = \left| \|g(x) - \mu_c(g)\|_2^2 - \|g(x) - \hat{\mu}_c(g)\|_2^2 \right|$$

$$= \|\mu_c(g) - \hat{\mu}_c(g)\|_2 \|2g(x) - (\mu_c(g) + \hat{\mu}_c(g))\|_2. \tag{23}$$

It was shown in (19) that $\|\mu_c(x)\|_2$ is upper bounded by $\sup_{x \in \mathcal{X}} \|g(x)\|_2$. One may similarly show the same for $\|\hat{\mu}_c(x)\|_2$:

$$\|\hat{\mu}_c(g)\|_2 = \left\| \frac{\frac{1}{n} \sum_{i=1}^n \phi_c(X_i) g(X_i)}{\frac{1}{n} \sum_{i=1}^n \phi_c(X_i)} \right\|_2 \leq \frac{\frac{1}{n} \sum_{i=1}^n \phi_c(X_i) \|g(X_i)\|_2}{\frac{1}{n} \sum_{i=1}^n \phi_c(X_i)}$$

$$\leq \frac{\sup_{x \in \mathcal{X}} \|g(x)\|_2 \frac{1}{n} \sum_{i=1}^n \phi_c(X_i)}{\frac{1}{n} \sum_{i=1}^n \phi_c(X_i)} = \sup_{x \in \mathcal{X}} \|g(x)\|_2.$$

As $g(x) \in [-L, +L]^d$, we have that $\|g(x)\|_2$, $\|\mu_c(x)\|_2$, and $\|\hat{\mu}_c(x)\|_2$ are all $L\sqrt{d}$-bounded, so

$$\left| d_g(x; c) - \hat{d}_g(x; c) \right| \leq 4L\sqrt{d} \|\mu_c(g) - \hat{\mu}_c(g)\|_2.$$

This together with (22) and (23) show that

$$\left| \Delta_n(g) - \hat{\Delta}_n(g) \right| \leq 8L\sqrt{d} \max_c \|\mu_c(g) - \hat{\mu}_c(g)\|_2. \tag{24}$$

Proposition 6, which we prove soon, upper bounds $\sup_{g \in \mathcal{G}} \|\mu_c(g) - \hat{\mu}_c(g)\|_2$. By a union bound argument over $c \in \{1, \ldots, k\}$, we get that for any fixed $\delta > 0$, there exists a constant $c_1 > 0$ such that

$$\sup_{g \in \mathcal{G}} \left| \Delta_n(g) - \hat{\Delta}_n(g) \right| \leq \frac{c_1 (8L\sqrt{d})}{\pi_{min}} \left[ \frac{d \mathcal{J}(2L)}{\sqrt{n}} + L \left( \sqrt{\frac{d \ln(k/\delta)}{n}} + \frac{\ln(dk/\delta)}{n} \right) \right],$$

with probability at least $1 - \delta$. □

This proposition upper bounds the supremum of the $\ell_2$ distance between cluster centres.

**Proposition 6.** *Suppose that Assumptions A1, A2, and A3 hold. Consider a fixed $c \in \{1, \ldots, k\}$. For any $\delta > 0$, there exists a constant $c_1 > 0$ such that with probability at least $1 - \delta$, we have*

$$\sup_{g \in \mathcal{G}} \|\mu_c(g) - \hat{\mu}_c(g)\|_2 \leq \frac{c_1}{\pi_{min}} \left[ \frac{d\mathcal{J}(2L)}{\sqrt{n}} + L \left( \sqrt{\frac{d \ln(1/\delta)}{n}} + \frac{\ln(d/\delta)}{n} \right) \right].$$

*Proof.* To shorten the formulae, we use the notation $\mathbb{E}_n \left[ f(X_i) \right] = \frac{1}{n} \sum_{i=1}^{n} f(X_i)$ to denote the empirical expectation. We can decompose $\mu_c(g) - \hat{\mu}_c(g)$ as follows

$$\mu_c(g) - \hat{\mu}_c(g) = \frac{\mathbb{E} \left[ \phi_c(X) g(X) \right]}{\mathbb{E} \left[ \phi_c(X) \right]} - \frac{\mathbb{E}_n \left[ \phi_c(X_i) g(X_i) \right]}{\mathbb{E}_n \left[ \phi_c(X_i) \right]}$$

$$= \frac{\mathbb{E} \left[ \phi_c(X) g(X) \right] - \mathbb{E}_n \left[ \phi_c(X_i) g(X_i) \right] + \mathbb{E}_n \left[ \phi_c(X_i) g(X_i) \right]}{\mathbb{E} \left[ \phi_c(X) \right]} - \frac{\mathbb{E}_n \left[ \phi_c(X_i) g(X_i) \right]}{\mathbb{E}_n \left[ \phi_c(X_i) \right]}$$

$$= \underbrace{\mathbb{E}_n \left[ \phi_c(X_i) g(X_i) \right] \left[ \frac{1}{\mathbb{E} \left[ \phi_c(X) \right]} - \frac{1}{\mathbb{E}_n \left[ \phi_c(X_i) \right]} \right]}_{\text{(I)}} + \underbrace{\frac{\mathbb{E} \left[ \phi_c(X) g(X) \right] - \mathbb{E}_n \left[ \phi_c(X_i) g(X_i) \right]}{\mathbb{E} \left[ \phi_c(X) \right]}}_{\text{(I)}}.$$

In the rest of the proof, we provide an upper bound for Term (I) and Term (II).

**Term (I):** Fix $\delta_1 > 0$. We denote $\mathbf{1}_d$ as the $d$-dimensional vector with all components equal to 1. As $g(x) \in [-L, +L]^d$ and $\phi_c(x) \geq 0$, we have

$$|\mathbb{E}_n \left[ \phi_c(X_i) g(X_i) \right]| \leq \frac{1}{n} \sum_{i=1}^{d} |\phi_c(X_i)||g(X_i)| \leq L\mathbf{1}_d \frac{1}{n} \sum_{i=1}^{d} |\phi_c(X_i)| = L\mathbf{1}_d \mathbb{E}_n \left[ \phi_c(X_i) \right],$$

where the comparison is dimension-wise.

Therefore,

$$\left| \frac{\mathbb{E}_n \left[ \phi_c(X_i) g(X_i) \right] \left( \mathbb{E}_n \left[ \phi_c(X_i) \right] - \mathbb{E} \left[ \phi_c(X) \right] \right)}{\mathbb{E}_n \left[ \phi_c(X_i) \right] \mathbb{E} \left[ \phi_c(X) \right]} \right| \leq \frac{L\mathbf{1}_d \mathbb{E}_n \left[ \phi_c(X_i) \right] \left| \mathbb{E}_n \left[ \phi_c(X_i) \right] - \mathbb{E} \left[ \phi_c(X) \right] \right|}{\mathbb{E}_n \left[ \phi_c(X_i) \right] \mathbb{E} \left[ \phi_c(X) \right]}$$

$$= \frac{L\mathbf{1}_d}{\mathbb{E} \left[ \phi_c(X) \right]} \left| \mathbb{E}_n \left[ \phi_c(X_i) \right] - \mathbb{E} \left[ \phi_c(X) \right] \right|. \quad (25)$$

We provide a probabilistic upper bound on $|\mathbb{E}_n \left[ \phi_c(X_i) \right] - \mathbb{E} \left[ \phi_c(X) \right]|$. Since $\phi_c$ is a fixed 1-bounded function, we use Hoeffding's inequality to upper bound it. After some manipulations, we obtain that

$$|\mathbb{E}_n \left[ \phi_c(X_i) \right] - \mathbb{E} \left[ \phi_c(X) \right]| \leq \sqrt{\frac{\ln(2/\delta_1)}{2n}},$$

with probability at least $1 - \delta_1$. We use this along (25) and the assumption that $\mathbb{E} \left[ \phi_c(X) \right] \geq \pi_{\min}$ to get

$$\|(\text{I})\|_2 \leq \frac{L\sqrt{d}}{\pi_{\min}} \sqrt{\frac{\ln(2/\delta_1)}{2n}}, \quad (26)$$

with probability at least $1 - \delta_1$.

**Term (II):** We want to upper bound the supremum of the norm of second term (II). Fix $\delta_2 > 0$. To simplify the notation, let us first define function $f(x) = \phi_c(x) g(x)$ corresponding to a function $g$. Functions $f$ are mapping from $\mathcal{X}$ to $\mathbb{R}^d$. Furthermore, we define the function space $\mathcal{F} = \{ x \mapsto \phi_c(x) g(x) : g \in \mathcal{G} \}$. For each dimension $s \in \{1, \ldots, d\}$, we also define $\mathcal{F}_s = \{ x \mapsto \phi_c(x) g_s(x) : g \in \mathcal{G} \}$, where $g_s$ is the $s$-th dimension of $g$. With this notation, we can write

$$\sup_{g \in \mathcal{G}} \left\| \frac{\mathbb{E} \left[ \phi_c(X) g(X) \right] - \mathbb{E}_n \left[ \phi_c(X_i) g(X_i) \right]}{\mathbb{E} \left[ \phi_c(X) \right]} \right\|_2 = \frac{1}{\mathbb{E} \left[ \phi_c(X) \right]} \sup_{f \in \mathcal{F}} \|\mathbb{E}_n \left[ f(X_i) \right] - \mathbb{E} \left[ f(X) \right]\|_2.$$

Let us focus on the supremum over $\mathcal{F}$ term. We have

$$\sup_{f \in \mathcal{F}} \|\mathbb{E}_n \left[ f(X_i) \right] - \mathbb{E} \left[ f(X) \right]\|_2^2 = \sup_{f \in \mathcal{F}} \sum_{s=1}^{d} |\mathbb{E}_n \left[ f_s(X_i) \right] - \mathbb{E} \left[ f_s(X) \right]|^2$$

$$\leq \sum_{s=1}^{d} \sup_{f_s \in \mathcal{F}_s} |\mathbb{E}_n \left[ f_s(X_i) \right] - \mathbb{E} \left[ f_s(X) \right]|^2.$$

We take the square root of both sides and use the fact that $\sqrt{a_1^2 + \ldots + a_d^2} \leq a_1 + \ldots + a_d$ (for non-negative $a_i$) to get

$$\sup_{f \in \mathcal{F}} \|\mathbb{E}_n[f(X_i)] - \mathbb{E}[f(X)]\|_2 \leq \sum_{s=1}^{d} \sup_{f_s \in \mathcal{F}_s} |\mathbb{E}_n[f_s(X_i)] - \mathbb{E}[f_s(X)]| .$$

Notice that as $\phi_c(x)$ is bounded by 1 and $g$ is $L$-bounded dimension-wise, each $f_s(x)$ is also $L$-bounded. We use Lemma 7 (Appendix B.3) on each term to obtain a high probability upper bound. We choose the parameters of that lemma as $\beta = 1$, $B = L$, $r = L^2$, and $\delta = \delta_2/d$. This leads to

$$\sup_{f \in \mathcal{F}} \|\mathbb{E}_n[f(X_i)] - \mathbb{E}[f(X)]\|_2 \leq \sum_{s=1}^{d} \left[ 4\mathbb{E}[R_n(\mathcal{F}_s)] + L\sqrt{\frac{2\ln(2d/\delta_2)}{n}} + \frac{8L}{3}\frac{\ln(2d/\delta_2)}{n} \right], \quad (27)$$

with probability at least $1 - \delta_2$.

In order to control $\mathbb{E}[R_n(\mathcal{F}_s)]$, we relate it to the covering number of $\mathcal{F}_s$. We see that the covering number of $\mathcal{F}_s$ can be upper bounded by the covering number of $\mathcal{F}$, which in turn can be upper bounded by the covering number of $\mathcal{G}$. First, we use Dudley's integral to get

$$\mathbb{E}[R_n(\mathcal{F}_s)] \leq \frac{4\sqrt{2}}{\sqrt{n}}\mathbb{E}\left[ \int_0^{\text{diam}(\mathcal{F}_s)} \sqrt{\log 2\mathcal{N}(u, \mathcal{F}_s, L_2(P_{X_{1:n}}))} \, du \right]. \quad (28)$$

The covering number argument is as follows. Choose any functions $f_s, f_s' \in \mathcal{F}_s$. For any sequence $x_{1:n}$, the squared of the empirical $\ell_2$-norm is

$$\frac{1}{n}\sum_{i=1}^{n} |f_s(x_i) - f_s'(x_i)|^2 \leq \frac{1}{n}\sum_{i=1}^{n}\sum_{r=1}^{d} |f_r(x_i) - f_r'(x_i)|^2 = \frac{1}{n}\sum_{i=1}^{n} \|f(x_i) - f'(x_i)\|_2^2$$

$$= \frac{1}{n}\sum_{i=1}^{n} \|\phi_c(x_i)g(x_i) - \phi_c(x_i)g'(x_i)\|_2^2 \leq \frac{1}{n}\sum_{i=1}^{n} \|g(x_i) - g'(x_i)\|_2^2 ,$$

where in the last step we used the fact that $\phi_c(x) \leq 1$.

An $\varepsilon$-cover of $\mathcal{G}$ w.r.t. $\|\cdot\|_{\infty,2}$ induces an $\varepsilon$-cover of $\mathcal{G}$ w.r.t. $L_2(P_{x_{1:n}})$, which in turn induces an $\varepsilon$-cover on $\mathcal{F}_s$ w.r.t. $L_2(P_{x_{1:n}})$, as the inequality above shows. Notice that this holds for any choice of $x_{1:n}$, including $X_{1:n}$ appearing in the Dudley's integral (28). Therefore, by (28) and setting $\text{diam}(\mathcal{F}_s) = 2L$, we have

$$\mathbb{E}[R_n(\mathcal{F}_s)] \leq \frac{4\sqrt{2}}{\sqrt{n}}\mathbb{E}\left[ \int_0^{2L} \sqrt{\log 2\mathcal{N}\left(u, \mathcal{G}, \|\cdot\|_{\infty,2}\right)} \, du \right] = \frac{4\sqrt{2}\mathcal{J}(2L)}{\sqrt{n}}, \quad (29)$$

where we used the definition of the entropy integral of $\mathcal{G}$ (9).

After plugging this upper bound in (27) and use the upper bound (26), which was obtained for Term (I), we see that

$$\sup_{g \in \mathcal{G}} \|\mu_c(g) - \hat{\mu}_c(g)\|_2 \leq \|(\text{I})\|_2 + \frac{1}{\mathbb{E}[\phi_c(X)]}\sup_{f \in \mathcal{F}} \|\mathbb{E}_n[f(X_i)] - \mathbb{E}[f(X)]\|_2$$

$$\leq \frac{L\sqrt{d}}{\pi_{\min}}\sqrt{\frac{\ln(2/\delta_1)}{2n}} +$$

$$\frac{d}{\pi_{\min}}\left[ \frac{16\sqrt{2}\mathcal{J}(2L)}{\sqrt{n}} + L\sqrt{\frac{2\ln(2d/\delta_2)}{n}} + \frac{8L}{3}\frac{\ln(2d/\delta_2)}{n} \right],$$

with probability at least $1 - (\delta_1 + \delta_2)$. We set $\delta_1 = \delta_2 = \delta/2$ and simplify the upper bound to obtain the desired result. $\qquad\square$

## B  AUXILIARY RESULTS

For the convenience of the reader, we collect some auxiliary definitions and results that are used in the our proofs.

### B.1    FUNCTION SPACE AND NORMS

Here we briefly define some of the notations that we use throughout the paper. Consider the domain $\mathcal{X}$ and a function space $\mathcal{F} : \mathcal{X} \to \mathbb{R}$. We do not deal with measure theoretic considerations, so we just assume that all functions involved are measurable w.r.t. an appropriate $\sigma$-algebra for that space. We use $\mathcal{M}(\mathcal{X})$ to refer to the set of all probability distributions defined over that space. We use symbols such as $\nu \in \mathcal{M}(\mathcal{X})$ to refer to probability distributions defined over that space.

We use $\|f\|_{p,\nu}$ to denote the $L_p(\nu)$-norm $(1 \leq p < \infty)$ of a measurable function $f : \mathcal{X} \to \mathbb{R}$, i.e.,

$$\|f\|_{p,\nu}^p \triangleq \int_{\mathcal{X}} |f(x)|^p \mathrm{d}\nu(x).$$

The supremum norm is defined as

$$\|f\|_\infty = \sup_{x \in \mathcal{X}} |f(x)|$$

Let $x_1, \ldots, x_n$ be a sequence of points in $\mathcal{X}$. We use $x_{1:n}$ to refer to this sequence. The empirical measure $P_n$ is the probability measure that puts a mass of $\frac{1}{n}$ at each $x_i$, i.e.,

$$P_n = P_{x_{1:n}} = \frac{1}{n} \sum_{i=1}^n \delta_{x_i},$$

where $\delta_x$ is the Dirac's delta function. For $\mathcal{D}_n = z_{1:x}$, the empirical $L_2(P_n)$-norm of function $f : \mathcal{X} \to \mathbb{R}$ is

$$\|f\|_{\mathcal{D}_n}^2 = \|f\|_{P_n}^2 = \frac{1}{n} \sum_{i=1}^n |f(x_i)|^2 .$$

We can also define other $L_p(P_n)$-norms similarly.[8] When there is no chance of confusion about $\mathcal{D}_n$, we may denote the empirical norm simply by $\|f\|_n$.

### B.2    THE METRIC ENTROPY AND THE COVERING NUMBER

We quote the definition of the covering number from Györfi et al. (2002).

**Definition 1** (Definition 9.3 of Györfi et al. 2002). *Let $\varepsilon > 0$, $\mathcal{F}$ be a set of real-valued functions defined on $\mathcal{X}$, and $\nu_{\mathcal{X}}$ be a probability measure on $\mathcal{X}$. Every finite collection of $N_\varepsilon = \{f_1, \ldots, f_{N_\varepsilon}\}$ defined on $\mathcal{X}$ with the property that for every $f \in \mathcal{F}$, there is a function $f' \in N_\varepsilon$ such that $\|f - f'\|_{p,\nu_{\mathcal{X}}} < \varepsilon$ is called an $\varepsilon$-cover of $\mathcal{F}$ w.r.t. $\|\cdot\|_{p,\nu_{\mathcal{X}}}$.*

*Let $\mathcal{N}(\varepsilon, \mathcal{F}, \|\cdot\|_{p,\nu_{\mathcal{X}}})$ be the size of the smallest $\varepsilon$-cover of $\mathcal{F}$ w.r.t. $\|\cdot\|_{p,\nu_{\mathcal{X}}}$. If no finite $\varepsilon$-cover exists, take $\mathcal{N}(\varepsilon, \mathcal{F}, \|\cdot\|_{p,\nu_{\mathcal{X}}}) = \infty$. Then $\mathcal{N}(\varepsilon, \mathcal{F}, \|\cdot\|_{p,\nu_{\mathcal{X}}})$ is called an $\varepsilon$-covering number of $\mathcal{F}$ and $\log \mathcal{N}(\varepsilon, \mathcal{F}, \|\cdot\|_{p,\nu_{\mathcal{X}}})$ is called the metric entropy of $\mathcal{F}$ w.r.t. the same norm.*

*Given a $x_{1:n} = (x_1, \ldots, x_n) \subset \mathcal{X}$ and its corresponding empirical measure $P_n = P_{x_{1:n}}$, we can define the empirical covering number of $\mathcal{F}$ w.r.t. the empirical norm $\|\cdot\|_{p,x_{1:n}}$ and is denoted by $\mathcal{N}_p(\varepsilon, \mathcal{F}, x_{1:n}) = \mathcal{N}(\varepsilon, \mathcal{F}, L_p(x_{1:n}))$.*

### B.3    RADEMACHER COMPLEXITY

We define Rademacher complexity and quote a result from Bartlett & Mendelson (2002). For more information about Rademacher complexity, we refer the reader to Bartlett & Mendelson (2002); Bartlett et al. (2005).

Let $\sigma_1, \ldots, \sigma_n$ be independent random variables with $\mathbb{P}\{\sigma_i = 1\} = \mathbb{P}\{\sigma_i = -1\} = 1/2$. For a function space $\mathcal{F} : \mathcal{X} \to \mathbb{R}$, define $R_n \mathcal{F} = \sup_{f \in \mathcal{F}} \frac{1}{n} \sum_{i=1}^n \sigma_i f(X_i)$ with $X_i \sim \nu$. The Rademacher complexity (or average) of $\mathcal{F}$ is $\mathbb{E}[R_n \mathcal{G}]$, in which the expectation is w.r.t. both $\sigma$ and $X_i$. Rademacher complexity appears in the analysis of the supremum of an empirical process right after the application of the symmetrization technique. As such, its behaviour is closely related to the

---

[8]Or maybe more clearly, $P_n(A) = \frac{1}{n} \sum_{i=1}^n \mathbb{I}\{x_i \in A\}$ for any measurable subset $A \subset \mathcal{X}$.

behaviour of the empirical process. One may interpret the Rademacher complexity as a complexity measure that quantifies the extent that a function from $\mathcal{F}$ can fit a noise sequence of length $n$ (Bartlett & Mendelson, 2002).

The following is a simplified (and slightly reworded) version of Theorem 2.1 of Bartlett et al. (2005).

**Lemma 7.** *Let $\mathcal{F} : \mathcal{X} \to \mathbb{R}$ be a measurable function space with $B$-bounded functions. Let $X_1, \ldots, X_n \in \mathcal{X}$ be independent random variables. Assume that for some $r > 0$, $\mathrm{Var}\,[f(X_i)] \leq r$ for every $f \in \mathcal{F}$. Then for every $\delta > 0$, with probability at least $1 - \delta$,*

$$\sup_{f \in \mathcal{F}} \left| \mathbb{E}\,[f(X)] - \frac{1}{n} \sum_{i=1}^{n} f(X_i) \right| \leq$$

$$\inf_{\beta > 0} \left\{ 2(1+\beta)\mathbb{E}\,[R_n(\mathcal{F})] + \sqrt{\frac{2r \ln(2/\delta)}{n}} + 2B \left( \frac{1}{3} + \frac{1}{\beta} \right) \frac{\log(2/\delta)}{n} \right\}.$$

# C    EXPERIMENTAL RESULTS

We give implementation details about the experiments of our submitted paper in Section C.1. We report the detailed performance of our model as a function of the output dimensionality and the number of hidden layers in the zero-shot learning context in Section C.2. We show in Section C.3 that our method can be generalized to hard clustering by using implicit centers. We give additional visualization results in Section C.4.

## C.1    IMPLEMENTATION DETAILS

We coded our method in PyTorch and ran all our experiments on a single Nvidia GeForce GTX 1060 which has 6GB of RAM.

PyTorch automatically calculates the gradient *w.r.t.* the mini-batch representation $F$. Nonetheless, it is worth mentioning that both the first and second arguments of our prediction function $\psi(F, M, \boldsymbol{\pi})$ depend on $F$ in the case where the centers are implicit (*i.e.*, when we write $M = \mathrm{diag}(Y^\top \mathbf{1}_n)^{-1} Y^\top F)$. In this case, the gradient of our loss function *w.r.t.* $F$ depends on both the first and second arguments of $\psi$.

### C.1.1    ZERO-SHOT LEARNING EXPERIMENTS

We now give details specific to the zero-shot experiments.

In the zero-shot learning experiment where $F$ and $M$ are computed from different sources (*i.e.*, images and text) and are the output of two different networks, the optimization is performed by alternately optimizing one variable while fixing the other.

**Mini-batch size:** The training datasets of CUB and Flowers contain 5894 and 5878 images, respectively. In order to fit into memory, we set our mini-batch sizes as 421 ($= 5894/14$) and 735 ($\approx 5878/8$) for CUB and Flowers, respectively.

Reed et al. (2016) and Snell et al. (2017) use 10 different views per image during training (middle, upper left, upper right, lower left and lower right crops for the original and horizontally-flipped image), and 1 view for test which is the middle crop of the original image. On the other hand, we use only 1 view per image (*i.e.*, the middle crop of the original image) during training and test.

**Optimizer:** We use the Adam optimizer with a learning rate of $10^{-5}$ to train both models $\varphi_{\theta_1}$ and $g_{\theta_2}$.

**Target soft assignment matrix:** When we use the representations provided by Snell et al. (2017), our target soft assignment matrix is $Y_1 = \boldsymbol{\psi}(\tilde{F}, \tilde{M}, \tilde{\boldsymbol{\pi}}) = \tilde{\Psi} \in \mathbb{Y}^{n \times k}$ where the matrices $\tilde{F}$ and $\tilde{M}$ are provided and $\tilde{\boldsymbol{\pi}} = \frac{1}{k}\mathbf{1}_k$. The elements of $\tilde{\Psi}$ are written $\tilde{\Psi}_{ic} = \frac{\tilde{\pi}_c \exp(-\mathsf{d}(\tilde{\mathbf{f}}_i, \tilde{\boldsymbol{\mu}}_c))}{\sum_{e=1}^{k} \tilde{\pi}_e \exp(-\mathsf{d}(\tilde{\mathbf{f}}_i, \tilde{\boldsymbol{\mu}}_e))}$. In this case, we formulate:

$$\mathsf{d}(\tilde{\mathbf{f}}_i, \tilde{\boldsymbol{\mu}}_c) = \frac{1}{10} \|\tilde{\mathbf{f}}_i - \tilde{\boldsymbol{\mu}}_c\|^2$$

Using a temperature of 10 made the optimization more stable as it avoided gradients with high values.

We use a temperature of 2 when using the representations provided by Reed et al. (2016).

**Initial temperature of our model:** To make our optimization framework stable, we start with a temperature of 50. We then formulate our Bregman divergence as:

$$d(\mathbf{f}_i, \boldsymbol{\mu}_c) = \frac{1}{50}\|\mathbf{f}_i - \boldsymbol{\mu}_c\|^2$$

where $\mathbf{f}_i$ and $\boldsymbol{\mu}_c$ are the representations learned by our model. We decrease our temperature by $10\%$ (*i.e.*, $\text{temp}_{t+1} = 0.9\text{temp}_t$) every 3000 epochs until the algorithm stops training.

We stop training at 10k epochs on CUB and 1k epochs on Flowers.

### C.1.2 VISUALIZATION EXPERIMENTS

We now give details specific to the visualization experiments.

**Dataset size:** To be comparable to t-SNE, we directly learn two-dimensional embeddings instead of neural networks. Our mini-batch size is the size of the test set (*i.e.*, the number of examples is $n = 10^4$ for most datasets except STL that contains $n = 8000$ test examples).

**Optimizer:** We use the RMSprop optimizer with a learning rate of $10^{-3}$, $\alpha = 0.99$, $\epsilon = 10^{-6}$, the weight decay and momentum are both 0, and the data is not centered. We also we formulate the empirical discrepancy loss $\Delta_n(\Psi, Y) = \sum_{i=1}^n D_{\mathsf{KL}}(\mathbf{y}_i\|\boldsymbol{\psi}_i)$ instead of $\Delta_n(\Psi, Y) = \frac{1}{n}\sum_{i=1}^n D_{\mathsf{KL}}(\mathbf{y}_i\|\boldsymbol{\psi}_i)$.

**Target soft assignment matrix:** Let us note $\mathbf{z}_i = [z_{i,1}, \cdots, z_{i,k}]^\top \in \mathbb{R}^k$ the vector containing the logits of the learned representation of the $i$-th test example. We formulate $\mathbf{y}_i = [y_{i,1}, \cdots, y_{i,k}]^\top \in \mathbb{R}^k$ our target assignment vector for the $i$-th test example as follows:

$$y_{i,c} = \frac{\exp(\frac{z_{i,c}}{\tau})}{\sum_{e=1}^k \exp(\frac{z_{i,e}}{\tau})} \tag{30}$$

where $\tau = 5$ for all the dataset except CIFAR-100 for which $\tau = 4$.

We report the quantitative scores for $\tau = 1$.

**Initial temperature of our model:** We learned our representation by using a fixed temperature of 1 (*i.e.*, using the standard squared Euclidean distance).

We stop the algorithm after 8000 iterations.

**Tuning t-SNE:** we tested different ranges of scaling (1/1, 1/10, 1/100) and perplexity (*i.e.*, 1, 10, 30 (default) and 100) and reported the representations that obtained the best quantitative results.

### C.2 IMPACT OF DIMENSIONALITY IN ZERO-SHOT LEARNING

Let $e \in \mathcal{N}$ be the dimensionality of the representations taken as input and $d$ the output dimensionality of the models $g_{\theta_1}$ and $g_{\theta_2}$, the architecture of the models is $e$-$d$ and $e$-$e$-$d$ in the 1 and 2 hidden layer cases, respectively. The hyperparameter $d \in \{16, 32, 64, \cdots, e\}$ is also a hyperparameter cross-validated on the validation set.

We give the detailed accuracy performance of our model in Tables 5 to 9.

• Table 5 reports the test performance of our model on CUB when using the features provided by Reed et al. (2016) as supervision for different numbers of hidden layers and values of output dimensionality of our model.

• Table 6 (resp. Table 7) reports the validation (resp. test) performance of our model on CUB when using the features provided by Snell et al. (2017) as supervision.

• Table 8 (resp. Table 9) reports the validation (resp. test) performance of our model on Flowers when using the features provided by Snell et al. (2017) as supervision.

Dimensionality reduction improves performance, though the optimal dimensionality is dataset specific. In general, increasing the number of hidden layers also helps.

| Dimensionality $d$ | 16 | 32 | 64 | 128 | 256 | 512 | 1024 |
|---|---|---|---|---|---|---|---|
| Linear model | 50.1 | 54.2 | 54.2 | 54.2 | 54.3 | 54.2 | 54.2 |
| 1 hidden layer | 51.4 | 54.4 | 54.6 | 54.3 | 54.5 | 54.7 | 53.8 |
| 2 hidden layers | 51.5 | 54.4 | 57.1 | 57.4 | 57.7 | **57.7** | 56.5 |

Table 5: Test accuracy (in %) as a function of $d$ when using DS-SJE (Reed et al., 2016) (Char CNN-RNN) as supervision on CUB

| Dimensionality $d$ | 16 | 32 | 64 | 128 | 256 | 512 |
|---|---|---|---|---|---|---|
| Linear model | 75.1 | 82.2 | 82.4 | 82.6 | 82.6 | 82.4 |
| 1 hidden layer | 77.0 | 81.9 | 82.1 | 82.4 | **83.3** | 82.9 |
| 2 hidden layers | 72.4 | 72.9 | 75.6 | 77.5 | 79.7 | 79.7 |

Table 6: Validation accuracy (in %) as a function of the output dimensionality $d$ when using ProtoNet (Snell et al., 2017) as supervision on CUB

| Dimensionality $d$ | 16 | 32 | 64 | 128 | 256 | 512 |
|---|---|---|---|---|---|---|
| Linear model | 56.0 | 58.3 | 58.6 | 58.6 | 58.6 | 58.4 |
| 1 hidden layer | 57.7 | 59.8 | 60.2 | 60.3 | **60.3** | 60.0 |
| 2 hidden layers | 57.4 | 58.5 | 59.4 | 59.6 | 59.5 | 59.3 |

Table 7: Test accuracy (in %) as a function of the output dimensionality $d$ when using ProtoNet (Snell et al., 2017) as supervision on CUB

| Dimensionality $d$ | 16 | 32 | 64 | 128 | 256 | 512 | 1024 |
|---|---|---|---|---|---|---|---|
| Linear model | 49.6 | 62.5 | 76.3 | 82.6 | 86.5 | 87.7 | 87.6 |
| 1 hidden layer | 86.7 | 87.1 | 87.1 | 87.3 | 87.7 | 87.9 | 87.8 |
| 2 hidden layers | 86.3 | 86.3 | 87.3 | 87.5 | 87.6 | **88.1** | 87.9 |

Table 8: Validation accuracy (in %) as a function of the output dimensionality when using ProtoNet (Snell et al., 2017) as supervision on Flowers

| Dimensionality $d$ | 16 | 32 | 64 | 128 | 256 | 512 | 1024 |
|---|---|---|---|---|---|---|---|
| Linear model | 54.6 | 58.0 | 61.5 | 64.2 | 64.2 | 64.7 | 64.4 |
| 1 hidden layer | 66.8 | 66.5 | 65.9 | 65.9 | 65.9 | 66.7 | 65.3 |
| 2 hidden layers | 67.2 | 67.3 | 67.9 | 67.7 | 67.7 | **68.2** | 66.0 |

Table 9: Test accuracy (in %) as a function of the output dimensionality when using ProtoNet (Snell et al., 2017) as supervision on Flowers

### C.3 GENERALIZATION TO HARD CLUSTERING

We validate that DRPR can be used to perform hard clustering as in Snell et al. (2017) but with implicit centers. To this end, we train a neural network with 2 convolutional layers on MNIST (LeCun et al., 1998) followed by a fully connected layer. Its output dimensionality is $d = 2$ or $d = 3$, the mini-batch size is $n = 1000$, the number of categories is $k = 10$ and the target hard assignment matrix $Y \in \{0, 1\}^{n \times k}$ contains category membership information (*i.e.*, $Y_{ic}$ is 1 if the example $\mathbf{f}_i$ belongs to category $c$, 0 otherwise). We train the model on the training set of MNIST and plot in Fig. 6 the representations of the test set. By assigning each test example to the category with closest centroid (obtained from the training set), the model obtains 98% (resp. 99%) accuracy when $d = 2$ (resp. $d = 3$). DRPR can then be learned for hard clustering when the centers are implicitly written as a function of the mini-batch matrix representation $F$ and the target hard assignment matrix $Y$.

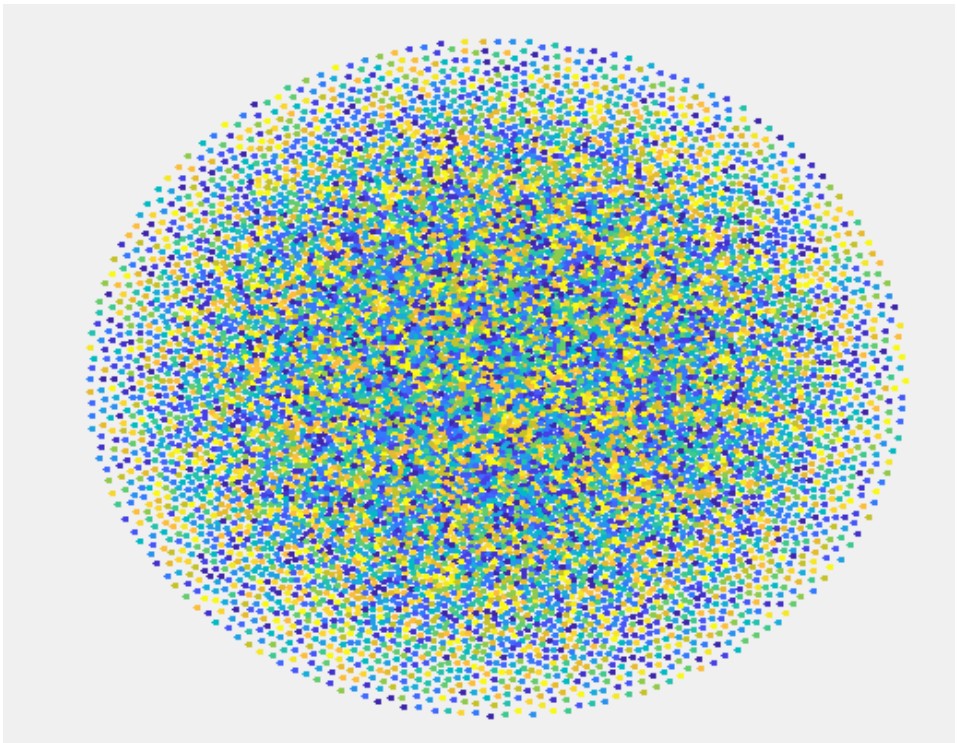

Figure 5: CIFAR 100 representation learned by t-SNE when using softmax representations as input

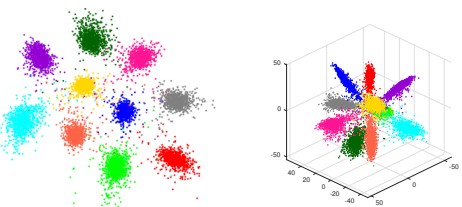

Figure 6: Visualization of the representation learned on MNIST by our approach in the supervised hard clustering setup. The left (resp. right) figure is the representation learned by our model when its output dimensionality is $d = 2$ (resp. $d = 3$).

## C.4 VISUALIZATION RESULTS

We now present visualization results.

### C.4.1 ARTIFACTS WITH T-SNE

Fig. 5 illustrates the CIFAR 100 representation learned by t-SNE when its input data is the target probability distribution that we give as supervision/input of our algorithm.

Following the recommendations mentioned in `https://lvdmaaten.github.io/tsne/` when the representation form a strange ball with uniformly distributed points and obtaining very low error, we decreased the perplexity from 30 (which is the value by default) to 10 and 1 and divided our data by 10, 100 and 1000. Nonetheless, we still obtained the same type of representation as in Fig. 5.

This kind of artifact is the reason why we only report results obtained with logits.

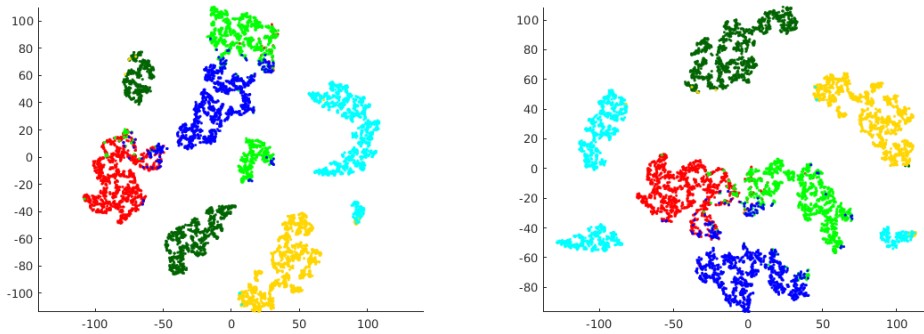

Figure 7: (left) Visualization obtained with t-SNE on the toy dataset when replacing the $\ell_2$ distance in the original space by the KL divergence to compare probability distribution representations. (right) Visualization obtained with t-SNE on the toy dataset when replacing the $\ell_2$ distance in the original space by the Jensen Shannon divergence.

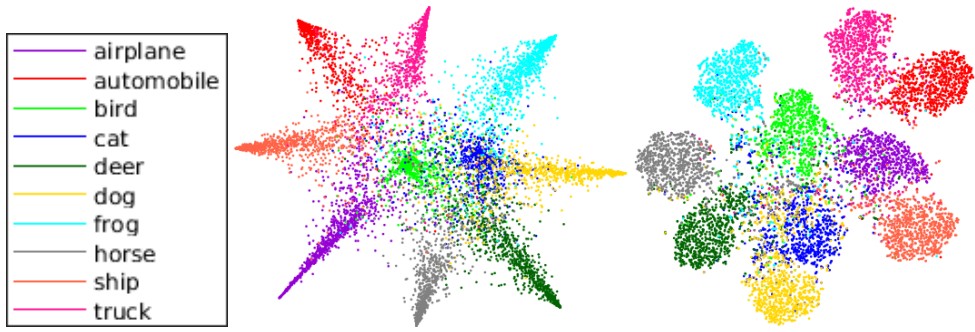

Figure 8: CIFAR 10 representation of DRPR and t-SNE

### C.4.2 ADAPTING T-SNE WITH OTHER DIVERGENCES

We plot in Fig. 7 the visualization obtained by t-SNE when using the KL or JS divergences to compare pairs of probability distribution representations. The representations obtained in this case are still worse than using the original 3-dimensional representations as the cluster structures are not preserved, nor the inter-cluster distances. This suggests that comparing pairs of examples, as done by t-SNE, is less appropriate than our method that considers similarities between examples and the different $k = 6$ clusters.

### C.4.3 ADDITIONAL RESULTS

Fig. 8 illustrates the DRPR and t-SNE representations of CIFAR 10. Animal categories are illustrated on the right whereas machines are on the left.

Fig. 9 illustrates the DRPR and t-SNE representations of CIFAR 100. We learned the representations by exploiting 100 clusters but plot only 20 colors (one for each superclass of CIFAR 100), which is why multiple spikes have the same color. Groups with same colors are better defined with our approach than with t-SNE, this means that different categories from the same superclass (*e.g.*, *hamster, mouse, rabbit, shrew, squirrel* which are *small mammals*) are grouped together with DRPR. One can observe a semantic structure in the 2D representation of DRPR: plants and insects are on the top left; animals are on the bottom left and categories on the right are outdoor categories. *Medium mammals* are also represented between *small mammals* and *large carnivores*.

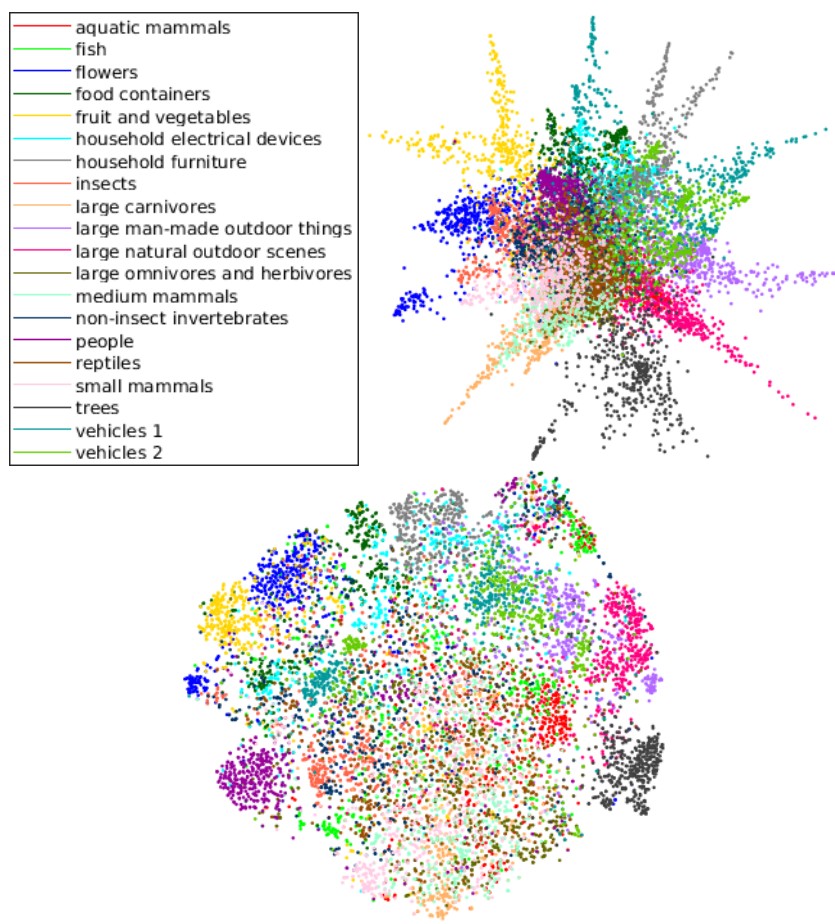

Figure 9: CIFAR 100 representation of DRPR and t-SNE

Figures 10, 11, 12 and 13 illustrate the representations learned by our model for the STL, MNIST, CIFAR 100 and CIFAR 10 datasets, respectively. Instead of using colors that represent the categories of the embeddings as done in the submitted paper, we directly plot the images.

In general, we observe that images towards the end of spikes consist of a clearly visible object in a standard viewpoint on a simple background. Those closer to the center often have objects with a non-standard viewpoint or have a complex textured background. At a high-level, the classes appear to be organized by their backgrounds.

Taking the STL-10 visualization as an example, *deer* and *horses* are close together since they both tend to be found in the presence of green vegetation. These classes are far from *boats* and *planes*, which often have solid blue backgrounds. Looking more closely, the ordering of classes is sensible. Planes are neighbors to both boats (similar background) and birds (similar silhouette). And trucks neighbor both cars (similar background) and horses, which appear visually similar, particularly for images in which the horse is pulling a cart.

Taking the MNIST visualization as another example, one can observe that written characters in spikes are easy to recognize as they correspond to examples for which the learned model has high confidence in its scores. On the other hand, ambiguous examples are between multiple spikes (*e.g.*, the characters *0* and *6* between spikes are more ambiguous than their neighbors in spikes).

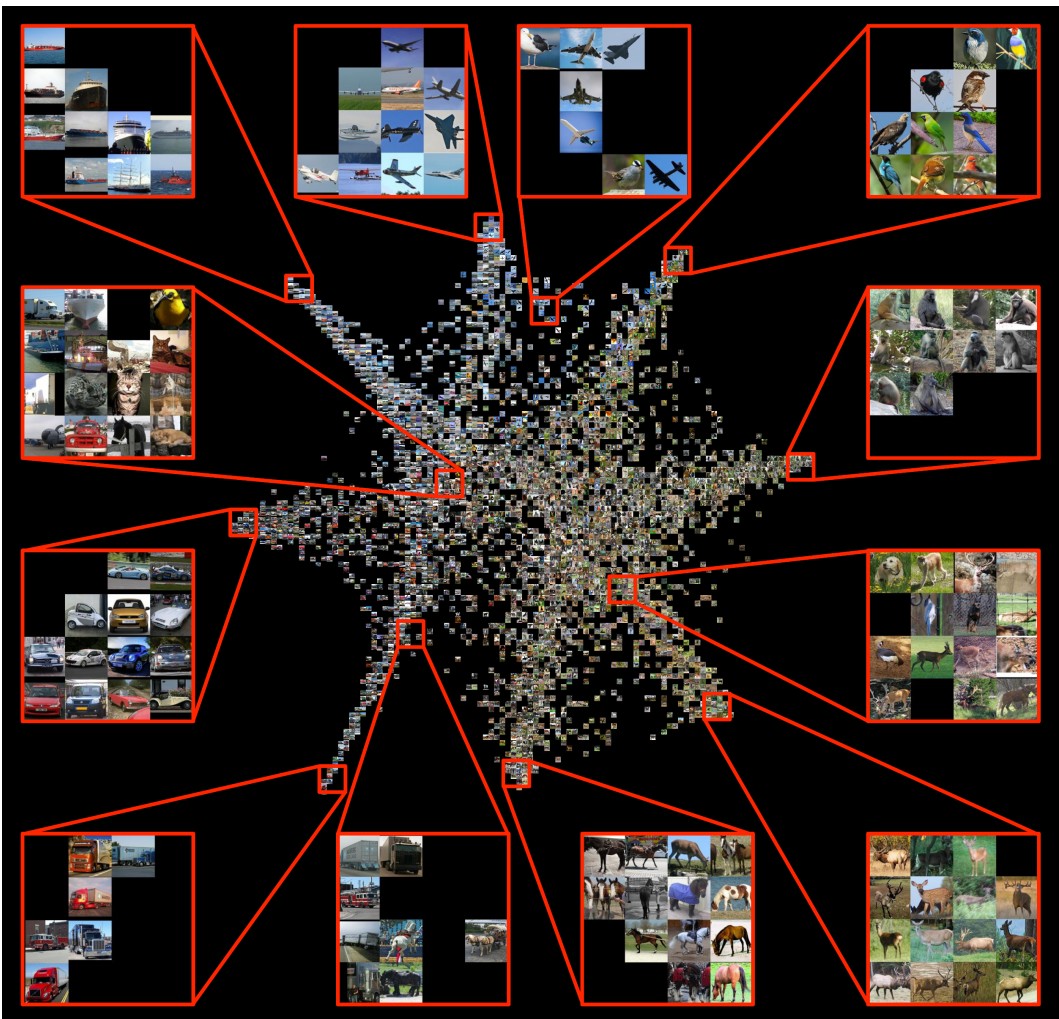

Figure 10: STL representation learned by DRPR.

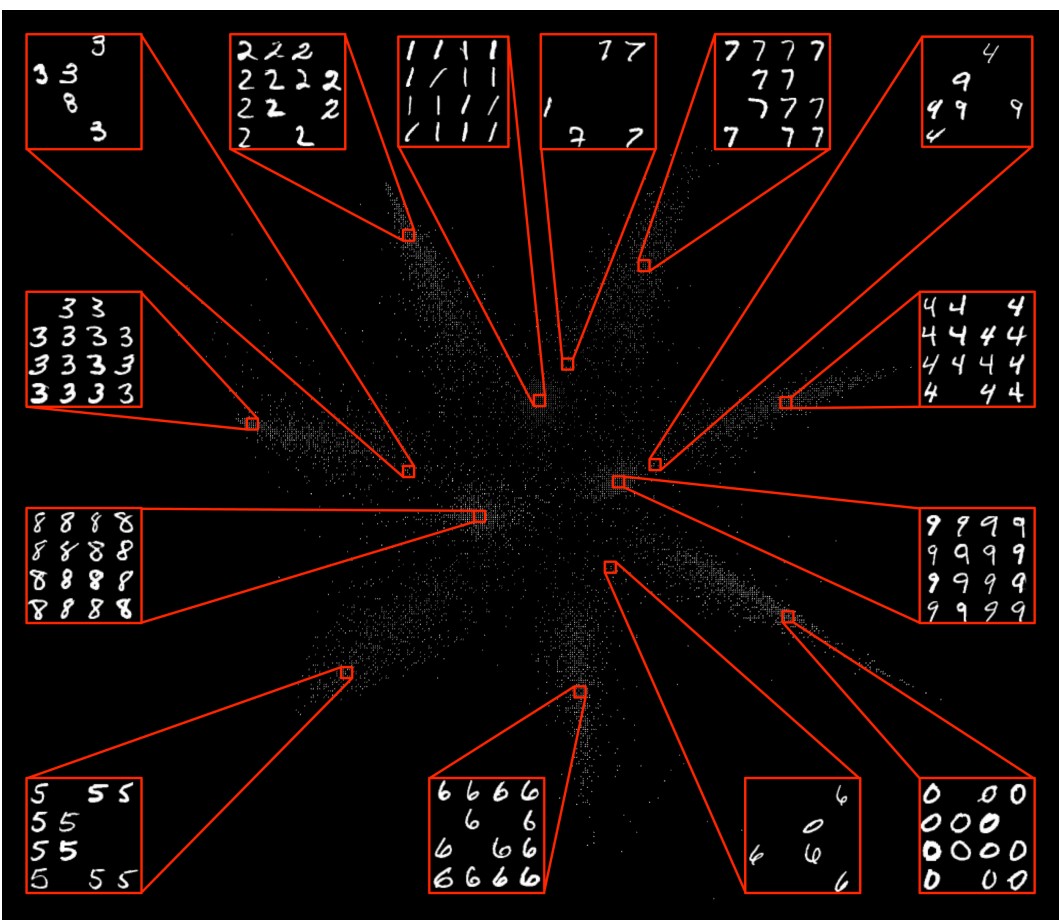

Figure 11: MNIST representation learned by DRPR.

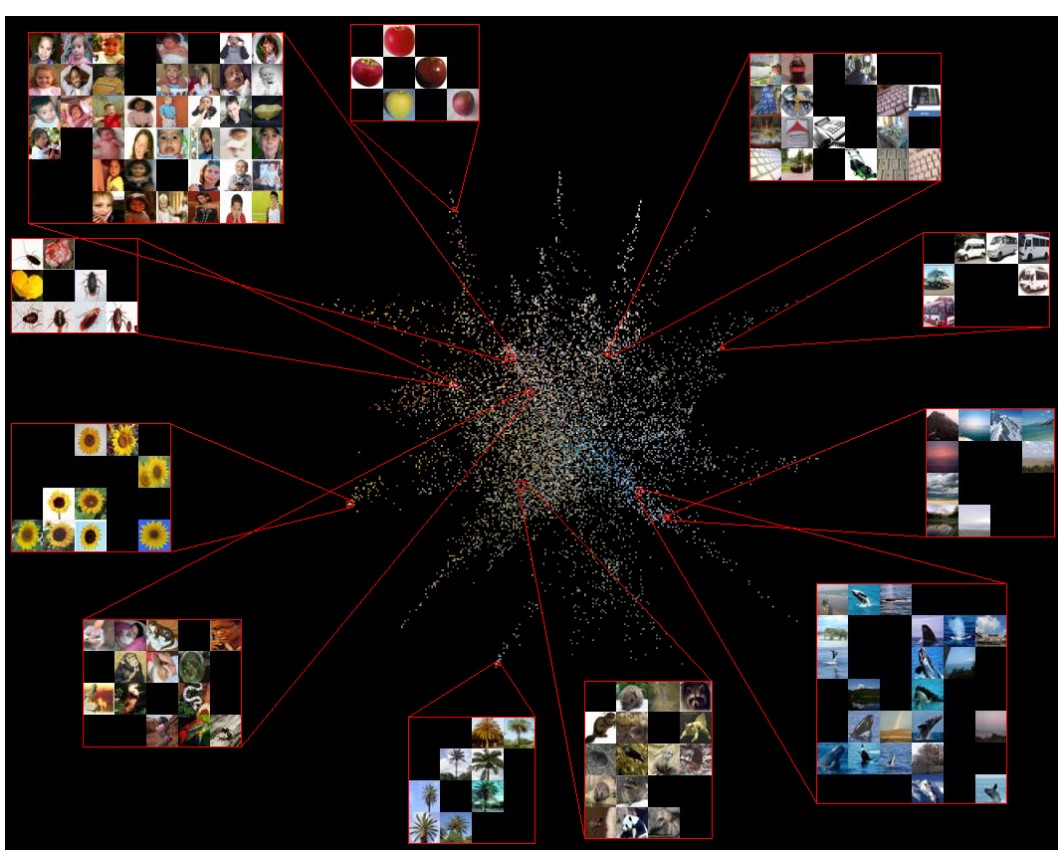

Figure 12: CIFAR 100 representation learned by DRPR.

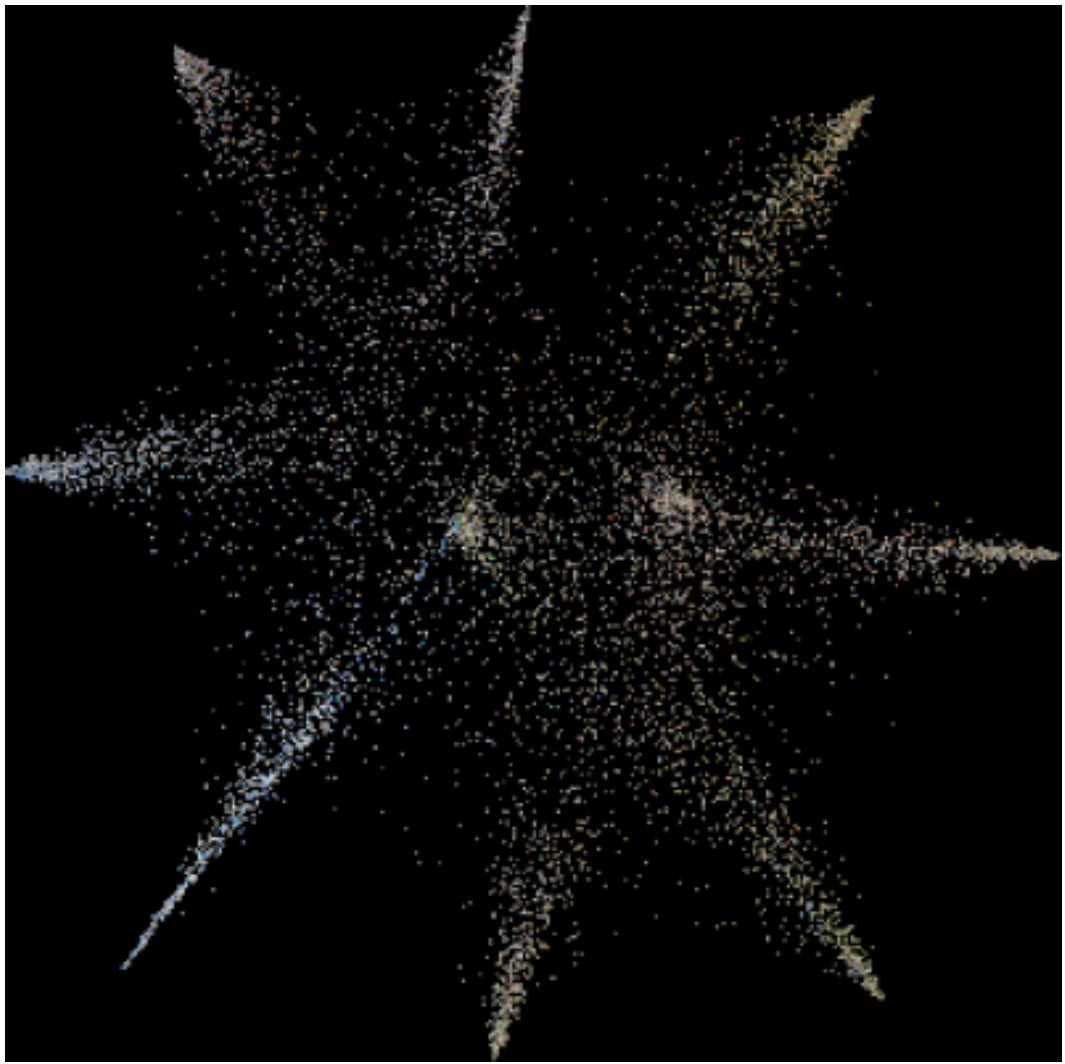

Figure 13: CIFAR 10 representation learned by DRPR.

