# OpenReview forum: "Dimensionality Reduction for Representing the Knowledge of Probabilistic Models"
_ICLR.cc/2019/Conference_

### Official Review · AnonReviewer2 · 2018-11-02
**Excellent paper with strong motivation, interesting proposed method, and comprehensive empirical results**

**Rating:** 9
**Confidence:** 3

**Review:**

Summary:

This paper introduces a new supervised dimensionality reduction model. Supervision is provided in the form of class probabilities and the learning algorithm learns low-dimensional representations such that posterior cluster assignment probabilities given the representations match the observed class probabilities. The representations can be learned directly or the parameters of a neural network can be learned which maps inputs to the lower-dimensional space. The authors provide an extensive theoretical analysis of the proposed method and evaluate it on dimensionality reduction, visualization, and zero-shot learning tasks.

Review:

Overall, I thought this was an excellent paper. The idea is well-motivated, the presentation is clear, and the evaluations are both comprehensive and provide insight into the behavior of the proposed methods (I will not comment on the theoretical analysis, as it is entirely contained in the supplemental materials). I was honestly impressed by the shear volume of content in this paper, particularly since I found none of it to be superfluous. Frankly, this paper might be better served as two papers or a longer journal paper, but that is hardly a reason not to accept it. I strongly recommend acceptance and have only a couple of comments on presentation.

Comments:

- When trying to understand the proposed method, I found it useful to expand out the full objective function and derive the gradients w.r.t. to f_i. If my maths were correct, the gradient of the objective w.r.t. f_i can be written as the difference between the expected gradient of the divergence w.r.t Y and the expected gradient of the divergence w.r.t. the posterior cluster assignment probabilities. Though not surprising in and of itself, the authors might consider including this equation as it really helped me understand what the learning algorithm was doing.

- The authors might consider adding a more complete description of the zero-shot learning task. My understanding of the task was that there are text descriptions of each category and at test time new text descriptions are added that were not in the training set. The goal is to map an unseen image to a class based on the text descriptions of the classes. A couple of sentences explaining this in the first paragraph of section 4.2 would help those who are not familiar with this zero-shot learning setup.

---

> ### Author Response · Authors · 2018-11-26
> **Response to AnonReviewer2**
>
> We thank you for your positive review and helpful suggestions. We address the comments in the following and have updated the paper accordingly:
>
> - “Comments: - When trying to understand the proposed method, I found it useful to expand out the full objective function and derive the gradients w.r.t. to f_i. If my maths were correct, the gradient of the objective w.r.t. f_i can be written as the difference between the expected gradient of the divergence w.r.t Y and the expected gradient of the divergence w.r.t. the posterior cluster assignment probabilities. Though not surprising in and of itself, the authors might consider including this equation as it really helped me understand what the learning algorithm was doing.”
>
> We thank the reviewer for this idea. As suggested, we have added the gradient of our optimization problem wrt the example f_i in the new Equation (5). To simplify its formulation, we consider that the matrix of centroids M does not depend on F (which is the case in the zero-shot learning task) and that priors are all equal. The gradient does depend on both Y and the posterior cluster assignment probabilities.
> More exactly, the magnitude of the gradient depends on both of theses scores (which means that a cluster with high score Y_ic will be given more importance).
> The gradient tries to make f_i closer to each centroid while separating it from all the centroids depending on their predicted scores as well.
>
> We have added a new paragraph called “Gradient interpretation” which discusses the gradient.
>
> - “The authors might consider adding a more complete description of the zero-shot learning task. My understanding of the task was that there are text descriptions of each category and at test time new text descriptions are added that were not in the training set. The goal is to map an unseen image to a class based on the text descriptions of the classes. A couple of sentences explaining this in the first paragraph of section 4.2 would help those who are not familiar with this zero-shot learning setup.”
>
> Thank you for this clear and succinct description of our zero-shot learning scenario. We have added this to the first paragraph of Section 4.2 as suggested.

---

### Official Review · AnonReviewer1 · 2018-11-03
**strong inductive bias of the model may not be appropriate for visualization**

**Rating:** 7
**Confidence:** 4

**Review:**

Authors propose a method of embedding training data examples into low-dimensional spaces such that mixture probabilities from a mixture model on these points are close to probability predictions from the original model in terms of KL divergence. Authors suggest two use-cases of such an approach: 1) data visualization, and 2) zero-shot learning. For the visualization use-case, authors compare against other dimensionality reduction methods with qualitative analysis on a synthetic problem, as well as evaluation metrics such as Neighborhood-Preservation Ratio and Clustering Distance Preservation Ratio. For zero-shot use-case, they take pre-trained models on two zero-shot tasks, and improve the accuracy by using probability outputs from pre-trained models as target.

Regarding the benefit of using the proposed method for visualization, the DRPR is making a strong assumption that representations of data points that belong to the same class form a uni-modal, Gaussian distribution (since authors don't experiment with distance functions other than L2). This inductive bias comes with a strong benefit when the assumption is true - as demonstrated in the toy dataset experiment - but when it is not true, the visualization would strongly distort the underlying structure of the model. And I don't believe this is a realistic assumption, because there has to be a reason that most deep-learning based classification models in the literature don't always use a model like (3) or Prototypical Networks instead of typical fully-connected + softmax layer, unless the data size is small and we need stronger inductive bias to improve the performance of the model.  That is, we usually don't think unimodality is the right assumption, even with learned representations. I suspect that the while DRPR might be good at visualizing relationships between class labels - especially which class can be easily confused with another - but would be worse at faithfully representing each data point, especially the ambiguity of class labels on individual ones. I would argue, however, that faithful representation of each data point is more important for scatter plots than relationship between classes, because the latter can be more effectively analyzed with other methods such as confusion matrices. As it is typical in most dimensionality reduction papers, I would encourage authors to consider more types of synthetic datasets which nonlinearity and multimodality are critical to be learned. I don't believe quantitative evaluation in Table 1 and 2 are very meaningful, because DRPR's objective function is much better aligned with these metrics than others.

Zero-shot experiments show a promising lift over the baseline pre-trained models. The kind of bias we should be careful about, however, is that when we distillate one model into another, the performance generally improves even when the same exact model is both the teacher and the student: (Furlanello et al, ICML 2018 https://arxiv.org/abs/1805.04770 ). Therefore, it would be interesting to compare against distillation with baseline models themselves.

Pros:
* Extensive theoretical and empirical analysis
* Simple idea that generalizes to multiple use-cases, which implies robustness of the approach as a methodology

Cons:
* Unimodal assumption is likely not realistic, which would result in misleading visualization of data
* Visualization analysis focuses on how class-relationships are preserved rather than faithful representation of each data point, which is a wrong target
* Synthetic experiment is conducted on a single, too simplistic one; more examples are needed to understand the capabilities of the model in more detail
* The bias of knowledge distillation is not controlled

---

> ### Author Response · Authors · 2018-11-26
> **Response to AnonReviewer1**
>
> We thank you for sharing your concerns about the paper. We will clarify your concerns regarding the assumptions made by DRPR and classification and distillation problems for zero-shot learning.
>
> - “ DRPR is making a strong assumption that representations of data points that belong to the same class form a unimodal distribution. I don't believe this is a realistic assumption.”
>
> While DRPR makes some assumptions on the distribution in the learned low-dimensional space, we would like to emphasize that DRPR makes no assumption on the input probability distribution in general.
> It is true that the toy dataset illustrates a special case where both the original and low-dimensional representations follow a unimodal Gaussian distribution (for each cluster).
> The goal of the toy experiment was to illustrate some weaknesses of t-SNE on a problem that is easy to visualize. We chose this toy dataset because it was easy to visualize the original 3D points themselves (input of t-SNE in Fig. 2 (b)), and also the soft assignment scores of the different points wrt the different clusters in the original space (input of t-SNE in Fig. 2 (c)). These soft probability scores are the target of our model.
>
> Our algorithm is similar to t-SNE in the sense that it assumes some distribution of the data in the low-dimensional space: t-SNE considers a Student-t distribution to compute similarity between pairs of points. Any kind of probability distribution can be used for the input space: t-SNE considers by default a conditional probability based on a Gaussian similarity between pairs of points, but any other kind of distribution can be given as input.
> DRPR is given target probability scores that can be computed from any distribution. In the zero-shot learning task, those scores indeed come from Gaussian mixtures. However, the targets in the visualization experiment come from neural networks trained with cross entropy and do not follow a Gaussian distribution
>
> - “Most deep-learning based classification models in the literature use softmax, we need stronger inductive bias to improve the performance of the model.”
>
> We agree that, in the usual classification task where the training and test categories are the same, learning a fully connected layer + softmax regression leads to state-of-the-art performance. However, the output dimensionality of the learned model is high and it is then difficult to interpret what has been learned by the model. Applying visualization techniques such as t-SNE has been proposed to interpret such complex models. Nonetheless, to the best of our knowledge, no visualization techniques exploit the fact that softmax classifiers have soft probabilistic interpretations.
>
> - “When we distillate one model into another, the performance generally improves even when the same exact model is both the teacher and the student (Furlanello et al., ICML 2018). Therefore, it would be interesting to compare against distillation with baseline models themselves.”
>
> We thank the reviewer for this suggestion. We have compared our method to two distillation strategies proposed in (Furlanello et al., ICML 2018) in the zero-shot learning task. We used it with the Prototypical Networks as the teacher since it obtains the best performance.
> In the first strategy, for each example, we preserve only the predicted category of the teacher model and convert the target as a one-hot vector. This actually corresponds to applying a second “layer” of prototypical network which is a special case of our method when the targets are hard assignments.
> In the second strategy, we preserve the predicted category and permute the scores of the category with lower scores.
> In both cases, the accuracy scores decreased relative to using only the Prototypical Network (without our method): about 55% on Birds and 60% on Flowers.
>
> This difference may be explained by the fact that (Furlanello et al., ICML 2018) consider a task where categories are the same during training and test. Applying wrong predictions based on a pre-trained teacher does not seem to affect accuracy.
> In our case, the set of training, validation and test categories are all different. Preserving relevant relative scores seems more crucial.
>
> - “Visualization analysis focuses on how class-relationships are preserved rather than faithful representation of each data point, which is a wrong target ”
>
> The “wrong” target depends on the task.
> Every dimensionality reduction framework has some criterion that is seeked to be optimized. PCA finds a linear transformation that maximizes the variance. t-SNE tries to preserve local neighborhood in both the high and low-dimensional spaces, which results in a poor preservation of inter-cluster distances. There is not a clear definition of “faithful representation” since it always depends on what is meant to be represented.
> The goal of DRPR is to find a low-dimensional space that best preserves the soft predicted scores of a classifier.

---

> > ### Comment · AnonReviewer1 · 2018-11-27
> > **good point distinguishing input distributions and target distributions**
> >
> > I admit that I was confusing input distributions with target distributions. Thanks for the clarification. Figure 4 confirms that data points need not be Gaussian shaped. I also mistakenly thought t-SNE plots on the right were DRPR plots before. Unimodality assumption of class distributions is still relevant, but I am not too concerned about it for two reasons: 1) t-SNE is not good at modeling multimodal distributions either both conceptually and empirically (ex: Figure 4), 2) it would be a straightforward to extend DRPR to allow multimodal distributions by allowing multiple centers to map to a single distributions. I will adjust my rating accordingly.

---

### Official Review · AnonReviewer3 · 2018-11-03
**What do we really minimize?**

**Rating:** 6
**Confidence:** 1

**Review:**

The paper deals with a problem formulation adjacent to that of the sufficient dimension reduction: given training set of pairs (x_i,y_i), how to reduce the dimension of the first element, i.e. map x_i --> f(x_i), so that f(x_i)'s still have all the information to recover y_i's.

In the paper, the output y_i is a probability distribution over k labels that softly describes inclusion of example i into k classes.

They consider a nonlinear case, i.e. the mapping f is taken from a prespecified set of mappings, parameterized by Theta (e.g. neural network). Then by "recovering y_i" they mean that EM algorithm for {f(x_i)} will result in a clustering of the data into k soft clusters similar to given {y_i}.

The algorithm that is presented is quite natural, though no guarantees that it will converge to something relevant were given. Theoretical analysis deals with a question --- how far the empirical discrepancy could be from the true expected one. Especially, easiness of substitution of \bar{Y}_{ic} with Y_{ic} in the algorithm is unclear (roughly speaking, the latter means that E-step is omitted in EM). If matrix Y in algorithm is fixed, why we need to compute \pi in the loop? Isn't it going to be the same? Does this algorithm really minimizes the discrepancy?

---

> ### Author Response · Authors · 2018-11-26
> **Response to AnonReviewer3**
>
> We would like to thank you for the feedback on the paper. We try to clarify your concerns regarding the convergence guarantee and why we need to compute prior at each iteration.
>
> - “The algorithm that is presented is quite natural, though no guarantees that it will converge to something relevant were given.”
> “Does this algorithm really minimize the discrepancy?”
>
> It is not clear what “something relevant” refers to, so we try to explain the result of Theorem 1 in the appendix. The theorem provides a finite-sample upper bound on the quality of the minimizer of the empirical loss, as defined in Eq. (6).
> The quality of the solution is measured according to the discrepancy (Delta_n), which is the expected KL-divergence between the teacher and the student. The teacher is defined by  \phi, and provides a distribution over k clusters.
> But since we only have access to n data points, the student minimizes the empirical discrepancy \Delta_n. The minimizer is \hat{g}, and it induces a distribution psi_\hat{g}.
> Theorem 1 shows that the expected KL-divergence between teacher \phi and student psi_\hat{g} is upper bounded by two terms:
> The best possible student within the function class G, from which \hat{g} is selected.
> Some estimation error terms that depend on the number of samples n and some properties of the function space G.
>
> We emphasize that our theoretical result does not concern the convergence of the optimization procedure, as noted in Footnote 5.
>
> A related question of the reviewer is that whether the algorithm really minimizes the discrepancy: The answer is that the algorithm minimizes the empirical version of the discrepancy. But the theory shows that doing so leads to a guarantee on the quality of resulting estimator, according to the true discrepancy (which has expectation, instead of a finite number of data points).
>
> - “Especially, easiness of substitution of \bar{Y}_{ic} with Y_{ic} in the algorithm is unclear (roughly speaking, the latter means that E-step is omitted in EM).”
>
> We thank the reviewer for pointing out this lack of clarity in our first submission. We assume that the reviewer refers to the statement “It is worth noting that we never apply the EM algorithm during training”. We do apply the E-step of the EM algorithm at each gradient descent iteration, but only once unlike the standard EM algorithm.
>
> More precisely, at each iteration, we know the optimal desired values of the M-step variables (i.e. the centroid matrix M and the prior vector \pi) as a function of the ground truth assignment matrix Y and the current representations of the mini-batch F. We then use these optimal values of M and \pi (formulated in step 4 of Algorithm 1) to compute the E-step, which corresponds to the predicted assignment matrix \Psi formulated in Eq. (3) (which you are denoting as \bar{Y}_{ic}).
> By definition of our optimization problem in Eq. (4), we minimize the average of KL divergence between the rows of Y and the rows of \Psi.
>
> We have updated the sentence accordingly.
>
> We would also like to emphasize that exploiting the target assignment matrix Y to compute the optimal M-step variables (i.e. centroid matrix) is commonly done in the supervised (hard) clustering literature [A,B].
>
> - “If matrix Y in algorithm is fixed, why we need to compute \pi in the loop? Isn't it going to be the same?”
>
> We think that you mean that if Y stays the same at every iteration, then \pi which depends only on Y should also stay the same at each iteration. This statement is correct.
> However, in the case where we train a neural network via mini-batches (e.g. in the zero-shot learning task or the hard clustering task in Section C.3), Y corresponds to the target assignment matrix of the mini-batch.
> Since each iteration then considers a different mini-batch F, the matrices Y and \pi then also change.
>
> Nonetheless, as mentioned in our paper, the priors can also be assumed to be all equal and then ignored.
> The matrix \pi can also be calculated according to the target assignments of the whole training dataset and then fixed.
>
>
> [A] Lajugie et al., Large-Margin Metric Learning for Constrained Partitioning Problems, ICML 2014
> [B] Law et al., Deep Spectral Clustering Learning, ICML 2017

---

### Meta-Review · Area_Chair1 · 2018-12-13
**Good paper**

**Confidence:** 4
**Recommendation:** Accept (Poster)

**Metareview:**

This paper introduces an approach for reducing the dimensionality of training data examples in a way that preserves information about soft target probabilistic representations provided by a teacher model, with applications such as zero-shot learning and distillation. The authors provide an extensive theoretical and empirical analysis, showing performance improvements in zero shot learning and finite sample error upper bounds. The reviewers generally agree this is a good paper that should be published.